

# Stochastic dissipative quantum spin chains (I): Quantum fluctuating discrete hydrodynamics

**Michel Bauer[1], Denis Bernard[2⋆] and Tony Jin[2⋆]**

**1** Institut de Physique Théorique de Saclay, CEA-Saclay & CNRS, 91191 Gif-sur-Yvette, France, and Département de mathématiques et applications, ENS-Paris, 75005 Paris, France
**2** Laboratoire de Physique Théorique de l'Ecole Normale Supérieure de Paris, CNRS, ENS & PSL Research University, UPMC & Sorbonne Universités, 75005 Paris, France

⋆ denis.bernard@ens.fr

## Abstract

Motivated by the search for a quantum analogue of the macroscopic fluctuation theory, we study quantum spin chains dissipatively coupled to quantum noise. The dynamical processes are encoded in quantum stochastic differential equations. They induce dissipative friction on the spin chain currents. We show that, as the friction becomes stronger, the noise induced dissipative effects localize the spin chain states on a slow mode manifold, and we determine the effective stochastic quantum dynamics of these slow modes. We illustrate this approach by studying the quantum stochastic Heisenberg spin chain.



## 1   Introduction

Non-equilibrium dynamics, classical and quantum, is one of the main current focuses of both theoretical and experimental condensed matter physics. In the classical theory, important theoretical progresses were recently achieved by solving simple paradigmatic models, such as the exclusion processes [1, 2]. This collection of results culminated in the formulation of the macroscropic fluctuation theory (MFT) [3] which provides a framework to study, and to understand, a large class of out-of-equilibrium classical systems. In the quantum theory, recent progresses arose through studies of simple, often integrable, out-of-equilibrium systems [4, 5]. Those deal for instance with quantum quenches [6–8], with boundary driven integrable spin chains [9, 10], or with transport phenomena in critical one dimensional systems either from a conformal field theory perspective [11–13] or from a hydrodynamic point of view [14, 15]. However, these simple systems generally exhibit a ballistic behaviour while the MFT deals with locally diffusive systems satisfying Fick's law. Therefore, to decipher what the quantum analogue of the macroscopic fluctuation theory could be –a framework that we may call the mesoscopic fluctuation theory–, we need, on the one hand, to quantize its set-up and, on the other hand, to add some degree of diffusiveness in the quantum systems under study.

The macroscopic fluctuation theory [3] provides rules for specifying current and density profile fluctuations in classical out-of-equilibrium systems. One of its formulation (in one dimension) starts from stochastic differential equations for the density $\mathfrak{n}(x,t)$ and the current $\mathfrak{j}(x,t)$, the first one being a conservation law:

$$\partial_t \mathfrak{n}(x,t) + \partial_x \mathfrak{j}(x,t) = 0, \tag{1}$$

$$\mathfrak{j}(x,t) = -D(\mathfrak{n})\partial_x \mathfrak{n}(x,t) + \sqrt{L^{-1}\sigma(\mathfrak{n})}\,\xi(x,t),$$

with $D(\mathfrak{n})$ the diffusion coefficient, $\sigma(\mathfrak{n})$ the conductivity and $\xi(x,t)$ a Gaussian space-time white noise, $\mathbb{E}[\xi(x,t)\xi(x',t')] = \delta(x-x')\delta(t-t')$. Here $L$ is the size of the system, so that the strength of the noise gets smaller as the system size increases. The statistical distribution of the noise $\xi(x,t)$ induces that of the density and of the current. The weakness of the noise for macroscopically large systems ensures that large deviation functions are computable through the solutions of extremization problems (which may nevertheless be difficult to solve). See refs. [3, 16–18] for instance.

The second equation in (1) is a constraint, expressing the current $\mathfrak{j}$ in terms of the density $\mathfrak{n}$ plus noise. A direct quantization of the evolution equations (1) seems difficult because of their diffusive nature and because, in a quantum theory, a constraint should be promoted to an operator identity. However, we can choose to upraise these two equations into dynamical

equations, of a dissipative nature, by adding current friction. For instance, we can lift these equations into the two following dynamical ones:

$$\partial_t \mathfrak{n} + \partial_x \mathfrak{j} = 0, \tag{2}$$
$$\tau_f \, \partial_t \mathfrak{j} = -D(\mathfrak{n})\partial_x \mathfrak{n} - \eta \mathfrak{j} + \sqrt{\eta L^{-1}\sigma(\mathfrak{n})}\, \xi,$$

where $\xi$ is again a space-time white noise. We have introduced a time scale $\tau_f$ to make these equations dimensionally correct and a dimensionless control parameter $\eta$, so that the current friction coefficient is $\eta\tau_f^{-1}$. In the large friction limit, $\eta \mathfrak{j} \gg \tau_f \partial_t \mathfrak{j}$, we recover the previous formulation. More precisely, let us rescale time by introducing a slow time variable $s = t/\eta$ and redefine accordingly the density $\hat{\mathfrak{n}}(x,s) = \mathfrak{n}(x,s\eta)$ and the current $\hat{\mathfrak{j}}(x,s) = \eta \mathfrak{j}(x,s\eta)$. By construction, these new slow fields satisfy the conservation law, $\partial_s \hat{\mathfrak{n}} + \partial_x \hat{\mathfrak{j}} = 0$, and the constraint $\hat{\mathfrak{j}} = -D(\hat{\mathfrak{n}})\partial_x \hat{\mathfrak{n}} + \sqrt{L^{-1}\sigma(\hat{\mathfrak{n}})}\, \hat{\xi}$, in the limit $\eta \to \infty$, (if $\eta^{-2}\partial_s \hat{\mathfrak{j}} \to 0$), with $\hat{\xi}(x,s) = \sqrt{\eta}\, \xi(x,s\eta)$ whose statistical distribution is identical to that of $\xi$.

In other words, the slow modes $\hat{\mathfrak{n}}$ and $\hat{\mathfrak{j}}$ of the dissipative dynamical equations (2), parametrized by the slow times $s = t/\eta$, satisfy the MFT equations (1), in the large friction limit. This is the strategy we are going to develop in the quantum case. Because equations (2) are first order differential equations (in time), they have a better chance to be quantizable. Quantizing these equations requires dealing with quantum noise. Fortunately, the notion of quantum stochastic differential equations exists and has been extensively developed in quantum optics [20] and in mathematics [19].

Quantum stochastic dissipative spin chains are obtained by coupling the quantum spin chain degrees of freedom to noise. The quantum evolution is then a random stochastic dissipative evolution. In the simplest case of the Heisenberg XXZ spin chain with random dephasing noise –the case we shall study in detail– the evolution of the spin chain density matrix $\rho_t$ is specified by a stochastic equation of the following form

$$d\rho_t = -i[h,\rho_t]\,dt - \frac{\eta \nu_f}{2}\sum_j [\sigma_j^z,[\sigma_j^z,\rho_t]]\,dt - \sqrt{\eta \nu_f}\sum_j i[\sigma_j^z,\rho_t]\,dB_t^j,$$

with $h$ the XXZ hamiltonian (whose definition is given below in eq.(17)) and $\sigma_j^z$ the spin half Pauli matrix on site $j$ of the chain. These are stochastic equations driven by real Brownian motions $B_t^j$, attached to each site of the spin chain. The drift terms include the XXZ unitary evolution plus a dissipative evolution inducing spin decoherence. They are of the Lindblad form. The noisy terms represent random dephasing, independently from site to site. They yield friction on the spin current, in a way similar to the classical theory described above. The dimensionless parameter $\eta$ controls the strength of the noise: the bigger is $\eta$, the stronger is the friction.

Motivated by the previous discussion, we look at the large friction limit $\eta \to \infty$. In this limit, the on-site random dephasing produces strong decoherence which induces a transmutation of the coherent hopping process generated by the XXZ hamiltonian into an incoherent jump process along the chain. This phenomena has two consequences. First, only a subset of observables survives in the large friction limit (at any fixed time) while the others vanish exponentially in this limit. Second, those remaining observables possess a slow dynamics (with respect to the slow time $s = t/\eta$) which codes for these random incoherent jump processes.

For instance, the effective slow dynamics for the spin observables reads

$$d\sigma_j^z = \frac{2\varepsilon^2}{\nu_f}\big(\sigma_{j-1}^z - 2\sigma_j^z + \sigma_{j+1}^z\big)ds + \sqrt{\frac{2\varepsilon^2}{\nu_f}}\big(d\mathbb{V}_s^j - d\mathbb{V}_s^{j-1}\big),$$

with $d\mathbb{V}_s^j$ noisy operators of a specific form, see eq.(32) for an explicit definition. It clearly describes a quantum, stochastic, diffusion process. The drift term is proportional to the discrete

Laplacian of the spins while the noise is the discrete gradient of random quantum operators so as to ensure local spin conservation. This equation codes for a mean diffusion (with constant diffusion constant) and for quantum and stochastic fluctuations through the random quantum operators $d\mathbb{V}_s^j$. It is worth comparing it with the classical noisy heat equation and with the classical MFT equations (1) recalled above.

Within a local hydrodynamic approximation discussed below, the above quantum stochastic equation can be mapped into a classical, stochastic, discrete hydrodynamic equation whose formal continuous limit coincides with the MFT equation (1). In other words, within this approximation, classical MFT is an appropriate description of these quantum, stochastic, systems. See eq.(36) below for more details.

This paper is organized as follows: In Section 2 we first define quantum stochastic versions of spin chains using tools from quantum noise theory. We then extract the relevant slow modes of those quantum stochastic systems and we describe their effective stochastic dynamics by taking the large friction limit of the previously defined quantum stochastic spin chain models. This general framework is illustrated in the case of the quantum stochastic Heisenberg XXZ spin chain in Section 3. In particular we describe how to take the large friction limit and how this limit leads to quantum fluctuating discrete hydrodynamic equations. A summary, extracting the main mechanism underlying this construction, as well as various perspectives, are presented in the concluding Section 4. We report most –if not all– detailed computations in six Appendices from A to F.

Let us point out that, when extracting the effective slow dynamics at large friction, we observe a Brownian transmutation –from real Brownian motions attached to the chain sites to complex Brownian motions attached to the links of the chain. Since we believe that this property has its own interest from a probability theory point of view, we make it mathematically precise in Appendix A. It may have applications in the study of the large noise limit of stochastic PDEs.

## 2 Quantum stochastic dissipative spin chains

To quantize the dynamical MFT equations (2) we are going to use quantum stochastic differential equations to couple a spin chain[1] to quantum noise. These are quantum analogues of Langevin equations. In our framework, there will be one quantum noise per lattice site in a way similar to the classical case in which there is one Brownian motion per position. The interaction between the spin chain and the quantum noise will be chosen appropriately to induce friction on the relevant spin currents.

### 2.1 Generalities

In a way similar to classical Langevin equations of the type considered in the introduction which codes for the interaction between a classical system with memoryless noise, quantum stochastic equations [19] code for the interaction between a quantum system and an infinitely large memoryless quantum reservoir representing quantum noise. Markovian quantum stochastic equations thus apply when memory effects in the reservoir are negligible [20].

Let us first give a brief description – using physical intuitions – of the origin of those processes and of the nature of the phenomena they are coding. No attempt to rigour or completeness has been made. These can be found in refs. [19–22] where more precise and detailed information can be found.

---

[1]Of course, we can formally extend this definition to higher dimensional lattices.

The simplest way to grasp what these equations encode consists in first considering a discrete analogue of quantum stochastic processes [21]. There, one considers a quantum system (in the present case, the spin chain), with Hilbert space $\mathscr{H}_{\text{sys}}$, and series of auxiliary quantum ancilla, each with quantum Hilbert space $\mathscr{H}_p$, so that the total Hilbert space is the tensor product $\mathscr{H}_{\text{sys}} \otimes \bigotimes_{n=1}^{\infty} \mathscr{H}_p$ (with an appropriate definition of the infinite tensor product). The infinite series of ancilla represents the quantum noise or the quantum coins, in a way similar to the coins to be drawn at random in order to define a random walk or a discrete stochastic process. Each of those ancilla is prepared in a given state and they interact successively and independently, one after the other, with the quantum system during a time laps of order say $\delta\tau$ (similarly to the way cavity QED experiments are performed [23]). Every time a new ancilla has interacted with the quantum system, the latter gets updated and entangled with that ancilla, in a way depending on the system-ancilla interaction. This iterative updating processes define the so-called discrete quantum stochastic evolutions.

By letting the time laps $\delta\tau$ to approach zero one gets the quantum stochastic equations (similarly to the way one gets the Brownian motion as a scaling limit of discrete random walk). When $\delta\tau \to 0$, the continuum of ancilla, indexed by the continuous time $t$, forms the quantum noise reservoir. In this limiting procedure, the (to be precisely defined) infinite product $\bigotimes_{n=1}^{\infty} \mathscr{H}_p$ becomes represented [21] by a (now well-defined) Fock space $\mathscr{H}_{\text{noise}}$ together with quantum noise operators $d\xi_t^j$ and $d\xi_t^{j\dagger}$, with canonical commutation relations $[d\xi_t^j, d\xi_t^{k\dagger}] = \delta^{j;k} dt$. Physically, these operators, when acting on the quantum noise Fock space, create and annihilate excitations on ancilla between time $t$ and $t+dt$ (but not on ancilla indexed by a time not in the interval $[t, t+dt]$). In the simplest case, the quantum noise satisfy the so-called quantum Itô rules,[2] $d\xi_t^{j\dagger} d\xi_t^k = 0$ and $d\xi_t^j d\xi_t^{k\dagger} = \delta^{j;k} dt$, as a consequence of vacuum expectation values in the Fock space.

In this limiting procedure, the discrete quantum stochastic evolutions yield continuous quantum flows of operators (in the Heisenberg picture) or density matrices (in the Schrödinger picture) on $\mathscr{H}_{\text{sys}} \otimes \mathscr{H}_{\text{noise}}$. The quantum ancilla, now indexed by the continuous time $t$, interact successively with the system. The flow between time $t$ and $t+dt$ is inherited from the updating of the system and its entanglement with the noise ancilla due to the system-noise interaction. Since the latter interaction was unitary in the discrete model, it remains so after the continuous limit has been taken. During a time interval $dt$ between time $t$ and $t+dt$, the unitary operator coding for this interaction can be written (by definition) as

$$\mathfrak{U}_{t+dt} \cdot \mathfrak{U}_t^{-1} = e^{-i\sqrt{\eta}\sum_j \left(e_j^\dagger d\xi_t^j + e_j d\xi_t^{j\dagger}\right)},$$

where the $e_j$'s are operators acting on $\mathscr{H}_{\text{sys}}$ and $d\xi_t^j$ are the quantum noise operators. The $e_j$'s code for the noise-system coupling. The fact that this unitary operator only involves $d\xi_t^j$ and their conjugates reflects the fact that, during this time interval, the system-noise interaction only involves the ancilla with time index between $t$ and $t+dt$. It expresses the absence of memory effects in this model of quantum noise.

The unitary operator $\mathfrak{U}_t$ codes for the system-noise interaction. The system may also be subject to his own evolution process, say defined via a Hamiltonian $h$ and a Lindbladian $L_s$. The flows $O \to O_t$ of any operator $O$ which combine the intrinsic system dynamics and system-noise interaction are then described by evolution equations of the form

$$dO_t = \left(i[h, O]_t + L_s^*(O)_t\right) dt + \eta L_b^*(O)_t\, dt + \sqrt{\eta} \sum_j \left(i[e_j^\dagger, O]_t\, d\xi_t^j + i[e_j, O]_t\, d\xi_t^{j\dagger}\right), \quad (3)$$

---

[2]Here we restrict ourselves to diagonal Itô rules but the generalization to the non diagonal case is simple. The formula we give in the text correspond to the so-called zero temperature quantum Itô rules but generalization to higher temperature is possible. see [19, 20, 22] for a brief introduction.

with $L_b$ a specific Lindblad operator induced by the system-noise interaction (to be described below). These are the so-called quantum stochastic differential equations. See e.g. [19–22] for more detailed information.

In the context of quantum stochastic spin chains, the index $j$ in $\xi_t^j$ labels the sites of the spin chain, the hamiltonian $h$ is that of the spin chain and the $L_s^*$ may come from an extra dissipative process acting on the spin chain in the absence of quantum noise. To specify the model we also have to declare how the spin chain degrees of freedom and the quantum noise are coupled by choosing the operators $e_j$. By convention, these are operators acting locally on the site $j$ of the spin chain. (But this choice can of course be generalized to operators $e_j$ acting on neighbour spins). We shall describe explicitly the example of the quantum stochastic XXZ Heisenberg spin in the following Section 3.

To simplify the discussion we shall now assume that the $e_j$'s are hermitian. Then, the quantum stochastic differential equations reduce to stochastic differential equations (SDE) with classical noise. They read:[3]

$$dO_t = \left(i[h,O]_t + L_s^*(O)_t\right)dt + \eta\, L_b^*(O)_t\, dt + \sqrt{\eta}\sum_j D_j^*(O)_t\, dB_t^j, \tag{4}$$

where $dB_t^j = d\xi_t^j + d\xi_t^{j\dagger}$ are classical Brownian motions normalized to $dB_t^j\, dB_t^k = \delta^{j;k}\, dt$. In this case, the derivatives $D_j^*$ and the Lindbladian $L_b^*$ are defined by $D_j^*(O) = i[e_j,O]$ and $L_b^*(O) = -\frac{1}{2}\sum_j[e_j,[e_j,O]]$, respectively. The evolution equations for density matrices are the dual of eq.(4). They read:

$$d\rho_t = \left(-i[h,\rho_t] + L_s(\rho_t)\right)dt + \eta\, L_b(\rho_t)\, dt + \sqrt{\eta}\sum_j D_j(\rho_t)\, dB_t^j, \tag{5}$$

with $D_j(\rho) = -i[e_j,\rho]$ and $L_b(\rho) = -\frac{1}{2}\sum_j[e_j,[e_j,\rho]]$.

If furthermore $L_s^* = 0$, still with the $e_j$'s hermitian, the flow (4) is actually a stochastic unitary evolution with infinitesimal unitary evolutions $U_{t+dt;t} = U_{t+dt}U_t^\dagger = e^{-idH_t}$ with hamilonian generators

$$dH_t = h\, dt + \sqrt{\eta}\sum_j e_j\, dB_t^j, \tag{6}$$

with $B_t^j$ normalized Brownian motions. The stochastic evolution equation for the operator $O_t$ reads $O_t \to O_{t+dt} = e^{+idH_t}O_t\, e^{-idH_t}$. The dual evolution equation for density matrices $\rho_t$ reads

$$\rho_t \to \rho_{t+dt} = e^{-idH_t}\rho_t\, e^{+idH_t}. \tag{7}$$

In this case, for each realization of the Brownian motions, the density matrix evolution is unitary, but its average (w.r.t. to the Brownian motions) is dissipative (encoded in a completely positive map).

## 2.2  Effective stochastic dynamics on slow modes

The dimensionless parameter $\eta$ controls the strength of the noise and the mean dissipation. As we argued in the Introduction, we aim at taking the large friction limit $\eta \to \infty$ in order to recover the quantum analogue of the macroscopic fluctuation theory (which we call the mesoscopic fluctuation theory).

The aim of this section is to describe a general enough step-up to deal with the large friction limit –which is also the strong noise limit– and determine the effective hydrodynamics of the slow modes in the limit of large dissipation $\eta \to \infty$. Since the aim is here to present a

---

[3]We use the Itô convention when writing classical stochastic differential equations.

possible framework, we will not enter into a detailed description of any peculiar models but only present the general logical lines. A more detailed and precise description will be provided in the following Section dealing with the stochastic Heisenberg XXZ model.

We first have to identify what the slow modes are? In the limit $\eta \to \infty$, the noise induced dissipation is so strong that all states $\rho_t$ are projected into states insensible to these dissipative processes. There is a large family –actually an infinite dimensional family in the example of the Heisenberg spin chain below– of such invariant states. These are the slow modes. They are parametrized by some coordinates –actually an infinite number of coordinates. The effective hydrodynamics is the dynamical evolution of these coordinates parametrized by the slow time $s = t/\eta$. (This is a slight abuse of language as we did not yet take the continuous space limit). In other words, the effective hydrodynamics is the dynamics induced on the slow mode manifold. It also describes the effective large time behaviour, which is dissipative and fluctuating by construction.

Let us first analyse the mean slow modes. Let $\bar\rho_t = \mathbb{E}[\rho_t]$ be the mean density matrix, where the expectation is with respect to the Brownian motions $B_t^j$. From eq.(5), it follows that its evolution equation is

$$d\bar\rho_t = \big(L(\bar\rho_t) + \eta L_b(\bar\rho_t)\big)dt, \tag{8}$$

where we set $L(\rho) = -i[h,\rho] + L_s(\rho)$. The maps $L$ and $L_b$ are operators, so-called super-operators, acting on density matrices. Since they are time independent, solutions of eq.(8) are of the form $\bar\rho_t = e^{t(L+\eta L_b)} \cdot \bar\rho_0$. Since $L_b$ and $L+\eta L_b$ are non-positive operators (by definition of a Lindblad operator), $\lim_{\eta\to\infty} e^{t(L+\eta L_b)} = \Pi_0$ with $\Pi_0$ the projection operator on $\mathrm{Ker}L_b$ which is composed of states such that $L_b(\rho) = 0$. In other word, $\lim_{\eta\to\infty} \bar\rho_t \in \mathrm{Ker}L_b$, and this forms the mean slow mode set. This projection mechanism of states on some invariant sub-space is analogous to the mechanism of reservoir engineering [24].

Since the space of mean slow modes is of large dimension, there is a remaining slow evolution. It can be determined via a perturbative expansion to second order in $\eta^{-1}$, as explained in Appendix D. It is of diffusive nature and it is parametrized by the slow time $s = t/\eta$. It reads (See the Appendix D for details, in particular we here assume that $\Pi_0 L \Pi_0 = 0$ as otherwise we would have to redefine the slow mode variables to absorb the fast motion generated by $\Pi_0 L \Pi_0$.):

$$\partial_s \hat{\bar\rho}_s = \mathfrak{A} \hat{\bar\rho}_s, \tag{9}$$

where $\mathfrak{A}$ is the super-operator, acting on density matrices in $\mathrm{Ker}L_b$, via

$$\mathfrak{A}\rho = -(\Pi_0 L (L_b^\perp)^{-1} L \Pi_0)(\rho), \tag{10}$$

with $\Pi_0$ the projector on $\mathrm{Ker}L_b$ and $(L_b^\perp)^{-1}$ the inverse of the restriction of $L_b$ on the (orthogonal) complement of $\mathrm{Ker}L_b$. Eq.(9) generates a diffusive flow on $\mathrm{Ker}L_b$, the mean slow mode manifold, which is diffusive and dissipative even if the original spin chain dynamics was not (i.e. even if $L_s = 0$ so that $L(\rho) = -i[h,\rho]$ is purely Hamiltonian). The effective slow diffusion is generated by the on-site noise.

Since the evolution $t \to \rho_t$ is stochastic there is more accessible information than the mean flow, and one may be willing to discuss the fluctuations and their large friction limit. A way to test this stochastic process is to look at expectations of any function $F(\rho_t)$ of the density matrix. For instance, we may consider polynomial functions, say $\mathrm{Tr}(O_1\rho_t)\cdots\mathrm{Tr}(O_p\rho_t)$, and look at their means, say $\mathbb{E}[\mathrm{Tr}(O_1\rho_t)\cdots\mathrm{Tr}(O_p\rho_t)]$. This amounts to look for statistical correlations between operator expectations. Let $\bar F_t := \mathbb{E}[F(\rho_t)]$ be the expectations (w.r.t. the Brownian motions $B_t^j$) of those functions. As for any stochastic process generated by Brownian motions, their evolutions are governed by a Fokker-Planck like equation of the form

$$\partial_t \mathbb{E}[F(\rho_t)] = \mathbb{E}[\mathfrak{D}F(\rho_t)] \tag{11}$$

with $\mathfrak{D}$ a second order differential operator (acting on functions of the random variable $\rho_t$). It decomposes into $\mathfrak{D} = \eta \mathfrak{D}_1 + \mathfrak{D}_0$ where $\eta \mathfrak{D}_1$ is the Fokker-Planck operator associated to the noisy dynamics and $\mathfrak{D}_0$ is the first order differential operator associated to the deterministic dynamics generated by the Lindbladian $L$.

Let us now identify what the slow mode observables are. Recall that these modes are those whose expectations are non trivial in the large friction limit $\eta \to \infty$. The formal solution of eq.(11) is

$$\mathbb{E}[F(\rho_t)] = \left( e^{t(\eta \mathfrak{D}_1 + \mathfrak{D}_0)} \cdot F \right)(\rho_0).$$

As a differential operator associated to a well-posed stochastic differential equation –that corresponding to the noisy part in eq.(4)–, the operator $\mathfrak{D}_1$ is non-positive. Hence the only observables which survive the large friction limit are the functions $F$ annihilated by $\mathfrak{D}_1$, i.e. such that $\mathfrak{D}_1 F = 0$. The functions which are not in the kernel of $\mathfrak{D}_1$ have expectations which decrease exponentially fast in time $t$ with a time scale of order $\eta^{-1}$.

The slow mode observables $F(\rho_t)$ are thus those in $\mathrm{Ker}\,\mathfrak{D}_1$. Their evolution –in the limit $\eta \to \infty$ at fixed $s = t/\eta$– can again be found by a perturbation theory to second order in $\eta^{-1}$. See Appendix E for details. The same formal manipulation as for the mean flow, but now dealing with operators acting on functions of the density matrix, tells us that the effective hydrodynamic equations are of the form

$$\partial_s \mathbb{E}[F(\rho_s)] = -\mathbb{E}\left[ \left( (\hat{\Pi}_0 \mathfrak{D}_0 (\mathfrak{D}_1^\perp)^{-1} \mathfrak{D}_0 \hat{\Pi}_0) \cdot F \right)(\rho_s) \right], \tag{12}$$

with $\hat{\Pi}_0$ be the projector on $\mathrm{Ker}\,\mathfrak{D}_1$ and $(\mathfrak{D}_1^\perp)^{-1}$ the inverse of the restriction of $\mathfrak{D}_1$ on the complement of its kernel. Again, as above, we made the simplifying hypothesis that $\hat{\Pi}_0 \mathfrak{D}_0 \hat{\Pi}_0 = 0$.

The above equation indirectly codes for the random flow on the slow modes. However, it may be not so easy, if not difficult, to make it explicit and tractable from this construction – although, it some case, such as in the XY model, it may be used to reconstruct the stochastic slow flow. So we now make a few extra hypothesis which will allow us to construct explicitly the stochastic flow on the slow modes.

Let us now suppose, that the initial stochastic dynamics $\rho_t \to \rho_{t+dt} = e^{-idH_t} \rho_t e^{+idH_t}$ is defined as in eq.(6) by the hamiltonian generator $dH_t = h\,dt + \sqrt{\eta} \sum_j e_j\,dB_t^j$ (i.e. we assume that $L_s^* = 0$ and $e_j^\dagger = e_j$ for all $j$). We furthermore assume that all local operators $e_j$ commute: $[e_j, e_k] = 0$. They generate commuting $U(1)$ actions. To simplify we furthermore assume that the hamiltonian $h$ has no $U(1)$-invariant component (this hypothesis is easily relaxed and will be relaxed in the case of the XXZ model). Under this hypothesis the noisy dynamics, generated by $\sqrt{\eta} \sum_j e_j\,dB_t^j$, can be explicitly integrated. It is simply the random unitary transformation $\tilde{U}_t = e^{-i\tilde{K}_t}$ with $\tilde{K}_t = \sqrt{\eta} \sum_j e_j B_t^j$. As a consequence, functions in $\mathrm{Ker}\,\mathfrak{D}_1$, which, by definition, are invariant under such unitary flows, are functions invariant under all $U(1)$s generated by the operators $e_j$. Hence, the slow mode observables are the functions $F$ invariant under all $U(1)$s,

$$F(\rho) = F(e^{+i\sum_j \theta_j e_j} \rho \, e^{-i\sum_j \theta_j e_j}), \tag{13}$$

for any real $\theta_j$'s.

The hydrodynamic flow is thus a flow a such invariant functions. But functions over a given space invariant under a group action are functions on the coset of that space by that group action. Hence, the slow mode observables are the functions on the coset space obtained by quotienting the space of system density matrices by all $U(1)$ actions generated by the operators $e_j$. Elements of this coset space are the fluctuating slow modes and the fluctuating slow hydrodynamic evolution takes place over this coset. These flows are defined up to gauge transformations. Indeed density matrices $\rho_t$ and $\tilde{\rho}_t = g_t \rho_t g_t^{-1}$, with $g_t$ some $U(1)$ transformations, represent the same elements of the coset space. If $\rho_t \to \rho_{t+dt} = e^{-id\hat{H}_t} \rho_t e^{+id\hat{H}_t}$

is the flow presented within the gauge $\rho_t$, the flow in the gauge transformed presentation is $\tilde{\rho}_t \to = e^{-id\tilde{H}_t} \tilde{\rho}_t e^{+id\tilde{H}_t}$ with gauge transformed hamiltonian $e^{-id\tilde{H}_t} = g_{t+dt} e^{-id\hat{H}_t} g_t^{-1}$. See Appendix B for the discussion of the simple toy model of a spin one-half illustrating this discussion.

To explicitly determine the fluctuating effective dynamics we use the opportunity that the noisy dynamics can be exactly integrated to change picture and use the interaction representation. Let us define the transformed density matrix $\hat{\rho}_t$ by

$$\hat{\rho}_t = e^{+i\sqrt{\eta}\sum_j e_j B_t^j} \rho_t \, e^{-i\sqrt{\eta}\sum_j e_j B_t^j}.$$

By construction, if $F$ is a $U(1)$-invariant function,

$$F(\rho_t) = F(\hat{\rho}_t),$$

so that we do not lose any information on the stochastic slow mode flow by looking at the time evolution of the transformed density matrix $\hat{\rho}_t$ (and this corresponds to a specific gauge choice). The latter is obtained from that of $\rho_t$ by going into the interaction representation via conjugacy, so that

$$\hat{\rho}_{t+dt} = e^{-id\hat{H}_t} \hat{\rho}_t \, e^{+id\hat{H}_t}, \tag{14}$$

with

$$d\hat{H}_t = e^{+i\sqrt{\eta}\sum_j e_j B_t^j} (h\,dt) \, e^{-i\sqrt{\eta}\sum_j e_j B_t^j}. \tag{15}$$

Going to the interaction representation allows us to extract most – if not all – of the rapidly oscillating phases which were present in the original density matrix evolution. Theses phases were making obstructions to the large friction limit and their destructive interferences were forcing the expectations of non $U(1)$s-invariant functions to vanish. Once these phases have been removed, it simply remains to show that the evolution equation (14) has a well-defined limit as a stochastic process. This is described in details in the case of the XXZ model in the following Section 3.

## 2.3 Remarks

Let us end this Section with a few remarks.

— In the above discussion, we made a few hypotheses in order to simplify the presentation. Some of them can be relaxed without difficulties. First we supposed that $\Pi_0 L \Pi_0 = 0$, with $\Pi_0$ the projector on $\mathrm{Ker} L_b$, or that $h$ has no $U(1)$s-invariant component. This hypothesis can be relaxed, in which case one has to modify slightly either the perturbation theory used to defined the slow dynamics or the unitary transformation defining the interaction representation. This is actually what we will have to do in the case of the XXZ model – and this is one of the main difference between the XXZ and the XY models. Second, when discussing the change of picture to the interaction representation we assume that $L_s = 0$. This can also be easily removed. The only difference will then be that the evolution equations in the interaction representation are not going to be random unitaries but random completely positive maps. Finally, in order to implement the map to the interaction representation we assume that the operators $e_j$'s were commuting. This is actually necessary as otherwise we would not be able to integrate the noisy dynamics and thus we would not be able to implement the unitary transformation mapping to the interaction representation.

— The noisy interaction, coded by the coupling $\sqrt{\eta}\sum_j e_j \, dB_t^j$, can be viewed as representing the interaction of local degrees of freedom with some local reservoir. This interaction has a tendency to force the system to locally relax towards local states invariant under the noisy interaction. For instance if we choose the operators $e_j$ to be proportional to the local energy density these local invariant states are locally Gibbs. So the noisy interaction can be seen as

enforcing some kind of local equilibrium or local thermalization if the noise-system is chosen appropriately. The typical relaxation time scale for these processes are proportional to $\eta^{-1}$ so that the large $\eta$ limit then corresponds to very fast local equilibration. The slow mode dynamics can then be interpreted as some kind of a fluctuating effective quantum hydrodynamics. Here and in the following, we are making a slight abuse of terminology as, usually, hydrodynamics refers to the effective dynamics of slow modes of low wave lengths. We are here going to describe slow mode dynamics without taking the small wave length limit (i.e. the slow mode dynamics on a discrete lattice space). We refer to this limit as discrete hydrodynamics.

— The construction of the effective slow stochastic dynamics we are presenting relies on analysing the hydrodynamic limit $\eta \to \infty$ at $s = t/\eta$ of the dynamics in the interaction representation. There, the hamiltonian generator is given by conjugating by the Brownian phase operator $e^{+i\sqrt{\eta}\sum_j e_j B_t^j}$ as made explicit in eq.(15). Implementing this conjugacy produces random phases of the following form (recall that $s = t/\eta$)

$$e^{i\sqrt{\eta}\sum_j \varphi_j B_t^j}\, dt \equiv_{\text{in law}} e^{i\eta \sum_j \varphi_j B_s^j}\, \eta ds, \tag{16}$$

with $\varphi_j$ real numbers. Here the equivalence relation refers to the fact that $B_{t=s\eta}^j = \sqrt{\eta} B_s^j$ in law. These phases are random, irregular and highly fluctuating in the limit of large friction. Our proof of the effective slow dynamics relies on the fact that, surprisingly, when $\eta \to \infty$, these phases converge to complex Brownian motions. We refer to this property as "Brownian transmutation". See Appendix A for a proof.

## 3  The stochastic quantum Heisenberg spin chain

We now illustrate the previous general framework in the simple, but non trivial, case of the XXZ Heisenberg spin chain. We first add noise to the usual Heisenberg spin chain model in a way to preserve the conservation law, as in the classical macroscopic fluctuation theory. We then describe the slow mode dynamics including its fluctuations and its stochasticity.

The XXZ local spins, at integer positions $j$ along the real line, are spin halves with Hilbert space $\mathbb{C}^2$. The XXZ Hamiltonian is a sum of local neighbour interactions, $h = \sum_j h_j$, with Hamiltonian density $h_j = \varepsilon(\sigma_j^x \sigma_{j+1}^x + \sigma_j^y \sigma_{j+1}^y + \Delta \sigma_j^z \sigma_{j+1}^z)$, so that:[4]

$$h = \varepsilon \sum_j (\sigma_j^x \sigma_{j+1}^x + \sigma_j^y \sigma_{j+1}^y + \Delta \sigma_j^z \sigma_{j+1}^z) = h^{xy} + \Delta h^{zz}, \tag{17}$$

where $\varepsilon$ fixes the energy scale and $\Delta$ is the so-called anisotropy parameter. We define $h^{xy} = \varepsilon \sum_j (\sigma_j^x \sigma_{j+1}^x + \sigma_j^y \sigma_{j+1}^y)$ and $h^{zz} = \varepsilon \sum_j \sigma_j^z \sigma_{j+1}^z$, so that $h = h^{xy} + \Delta h^{zz}$. Let $n_j = \frac{1}{2}(1 + \sigma_j^z)$ be the local density and $J_j = \varepsilon(\sigma_j^x \sigma_{j+1}^y - \sigma_j^y \sigma_{j+1}^x)$ be the local current. The equations of motion, $\partial_t O = i[h, O]_t$, are:

$$\partial_t n_j = J_{j-1} - J_j,$$
$$\partial_t J_j = K_j - K_{j+1} - \Delta G_j,$$

with

$$K_j = 2\varepsilon^2\big(2\sigma_j^z + (\sigma_{j-1}^x \sigma_j^z \sigma_{j+1}^x + \sigma_{j-1}^y \sigma_j^z \sigma_{j+1}^y)\big)$$

and

$$G_j = 2\varepsilon^2(\sigma_{j-1}^z - \sigma_{j+2}^z)(\sigma_j^x \sigma_{j+1}^y + \sigma_j^y \sigma_{j+1}^x).$$

---

[4]Here $\sigma^{x,y,z}$ are the standard Pauli matrices, normalized to $(\sigma^{x,y,z})^2 = 1$, with commutation relations $[\sigma^x, \sigma^y] = 2i\sigma^z$ and cyclic permutations. As usual, let $\sigma^{\pm} = \frac{1}{2}(\sigma^x \pm i\sigma^y)$. They satisfy $\sigma^z \sigma^{\pm} = -\sigma^{\pm} \sigma^z = \pm\sigma^{\pm}$ and $\sigma^+ \sigma^- = \frac{1}{2}(1 + \sigma^z)$ and $\sigma^- \sigma^+ = \frac{1}{2}(1 - \sigma^z)$.

The first equation is a conservation law, the second codes for spin wave propagation. The simple case of the XY model, corresponding to $\Delta = 0$, is described at the end of this Section.

## 3.1 The stochastic XXZ model

We now add noise and write the quantum stochastic equation in such way as to preserve the conservation law. This completely fixes the form of the quantum SDE. Indeed, demanding that the conservation law $dn_j = (J_{j-1} - J_j)dt$ holds in presence of quantum noises imposes that all $e_j$'s commute with $\sigma_j^z$ and hence demands that $e_j \propto \sigma_j^z$ (if the proportionality coefficients are complex we absorb the phases into a redefinition of the noise). Thus, we set $e_j = \sqrt{\nu_f}\,\sigma_j^z$ where the coefficient $\nu_f$, with the dimension of a frequency (inverse of time), is going to be interpreted as the friction coefficient. The quantum SDE, defining the quantum stochastic Heisenberg XXZ model, is then

$$dO_t = i[h, O]_t\, dt + \eta L_b(O)_t\, dt + \sqrt{\eta} \sum_j D_j^*(O)_t\, dB_t^j, \tag{18}$$

with $D_j^*(O) = i\sqrt{\nu_f}[\sigma_j^z, O]$ and $L_b(O) = -\frac{\nu_f}{2}\sum_j[\sigma_j^z,[\sigma_j^z, O]]$. Again we have introduced a control dimensionless parameter $\eta$. Because of the remarkable relations $L_b(n_j) = 0$ and $L_b(J_j) = -4\nu_f J_j$, the equations of motion (18) for the density $n_j$ and the current $J_j$ are:

$$dn_j = (J_{j-1} - J_j)dt,$$
$$dJ_j = (K_j - K_{j+1} - \Delta G_j - 4\eta\nu_f J_j)dt + 2\sqrt{\eta\nu_f}\, h_j^{xy}(dB_t^{j+1} - dB_t^j), \tag{19}$$

with $h_j^{xy} = \varepsilon(\sigma_j^x\sigma_{j+1}^x + \sigma_j^y\sigma_{j+1}^y)$.

As pointed out in Section 2, this quantum SDE is actually a stochastic unitary evolution $O \to O_t = U_t^\dagger O U_t$ with

$$U_{t+dt}U_t^\dagger = e^{-idH_t}, \quad dH_t = h\, dt + \sqrt{\eta\nu_f}\sum_j \sigma_j^z\, dB_t^j. \tag{20}$$

The evolution equation for the density matrix, $\rho_t \to \rho_{t+dt} = e^{-idH_t}\rho_t e^{+idH_t}$, can be written in the following form (which we may call a "stochastic Lindblad equation"):

$$d\rho_t = -i[h, \rho_t]\, dt + \eta L_b(\rho_t)\, dt + \sqrt{\eta}\sum_j D_j(\rho_t)\, dB_t^j, \tag{21}$$

with, again, $D_j(\rho) = -i\sqrt{\nu_f}[\sigma_j^z, \rho]$ and $L_b(\rho) = -\frac{\nu_f}{2}\sum_j[\sigma_j^z,[\sigma_j^z, \rho]]$. Let us insist that, for each realization of the Brownian motions, the density matrix evolution is unitary, but its mean (w.r.t. to the Brownian motions) is dissipative.

This model can of course be generalized by including inhomogeneities. This amounts to replace the hamiltonian generator $dH_t = h\, dt + \sqrt{\eta\nu_f}\sum_j \sigma_j^z dB_t^j$ by $dH_t = h\, dt + \sqrt{\eta\nu_f}\sum_j \kappa_j \sigma_j^z dB_t^j$, with $\kappa_j$ real numbers controlling the strength of the noise independently on each site.

We could also have introduced variants of the model by changing the coupling between the noise and the spin chain degrees of freedom. Besides the previous definition another natural choice would have been to couple the noise and the spin chain via the local energy density instead of the local magnetization density. The hamiltonian would then have been $dH_t = h\, dt + \sqrt{\eta\nu_f}\sum_j \kappa_j h_j dB_t^j$ with $h_j$ the local energy density. But this model is more difficult to solve because the $h_j$'s do not commute.

To simplify the following discussion we restrict ourselves to the simple homogeneous $\sigma_j^z$ coupling. In order to ease the reading, we repeat some of the general argument presented in the previous Section – even though this may induce a few (tiny) repetitions.

### 3.2 The mean diffusive dynamics of the stochastic XXZ model

Let us start by discussing the mean dynamics and its large friction limit. The equations of motion for the mean density $\bar{n}_j = \mathbb{E}[n_j]$ and the mean current $\bar{J}_j = \mathbb{E}[J_j]$ are the following dissipative equations:

$$\partial_t \bar{n}_j = \bar{J}_{j-1} - \bar{J}_j,$$
$$\partial_t \bar{J}_j = \bar{K}_j - \bar{K}_{j+1} - \Delta \bar{G}_j - 4\eta \nu_f \bar{J}_j,$$

with $\bar{K}_j = \mathbb{E}[K_j]$ and $\bar{G}_j = \mathbb{E}[G_j]$. Their structures are similar to those of the classical MFT, see eq.(2). The dissipative processes coded by the Lindbladian $L_b$ effectively induce current friction with a friction coefficient proportional to $\nu_f$. The formal large friction limit, $\eta \to \infty$, imposes the operator constraint $4\eta \nu_f \bar{J}_j \simeq \bar{K}_j - \bar{K}_{j+1} - \Delta \bar{G}_j$, which may be thought as a possible quantum analogue of Fick's law. Of course they do not form a closed set of equations.

The mean density matrix $\bar{\rho}_t = \mathbb{E}[\rho_t]$ evolves dissipatively through $d\bar{\rho}_t = \left(-i[h, \bar{\rho}_t] + \eta L_b(\bar{\rho}_t)\right)dt$, or explicitly

$$d\bar{\rho}_t = -i[h, \bar{\rho}_t] dt - \eta \frac{\nu_f}{2} \sum_j [\sigma_j^z, [\sigma_j^z, \bar{\rho}_t]] dt.$$

The mean dynamics has been studied in ref. [25]. The unique steady state, which is reached at infinite time, is the uniform equilibrium state proportional to $e^{-\mu \sum_j \sigma_j^z}$. The effective hydrodynamics, i.e. the limit $\eta \to \infty$ at $s = t/\eta$ fixed, describes how this equilibrium state is attained asymptotically.

At large $\eta$ the mean flow is dominated by the noisy dissipative processes generated by $\eta L_b$. It converges to locally $L_b$-invariant states, i.e. to states in Ker$L_b$, because the relaxation time for this dissipative process is proportional to $\eta^{-1}$. The Lindbladian $L_b$ is a sum of local terms, $L_b = \sum_j L_b^j$ with $L_b^j(\rho) = -\frac{\nu_f}{2}[\sigma_j^z, [\sigma_j^z, \rho]]$. The $L_b^j$'s commute among themselves and are all negative operators. The kernel of each $L_b^j$ are spanned by the identity and $\sigma_j^z$. Thus the locally $L_b$-invariant states are the density matrices with local components diagonal in the $\sigma_j^z$'s basis. For instance, if we assume factorization, they are of the form $\otimes_j \frac{1}{2}(1 + \bar{S}_j \sigma_j^z)$. But a general locally $L_b$-invariant state may not be factorized. These are the mean slow modes. Let us denote them $\hat{\bar{\rho}}$.

As explained above in Section 2.2, since the mean slow modes form a high dimensional manifold they undergo a slow dynamical evolution (w.r.t. to the slow time $s = t/\eta$). This mean slow dynamics is determined via a second order perturbation theory. It reads $\partial_s \hat{\bar{\rho}}_s = \mathfrak{A}\hat{\bar{\rho}}_s$ with $\mathfrak{A}(\rho) = -(\Pi_0 L (L_b^{\perp})^{-1} L \Pi_0)(\rho)$ for $\rho \in \text{Ker}L_b$, with $L(\rho) = -i[h, \rho]$ and $(L_b^{\perp})^{-1}$ the inverse of the restriction of $L_b$ to the complement of Ker$L_b$ and $\Pi_0$ the projector on Ker$L_b$. Peculiar properties of the space of operators, of the Heisenberg hamiltonian, and especially of the Lindbladian $L_b$, allow us to show that, in this particular case, the operator $\mathfrak{A}$ simplifies to:

$$\partial_s \hat{\bar{\rho}}_s = \mathfrak{A}\hat{\bar{\rho}}_s = -\frac{1}{4\nu_f} \Pi_0[h, [h, \hat{\bar{\rho}}_s]]. \tag{22}$$

or equivalently, thanks to the specific form of $h$,

$$\partial_s \hat{\bar{\rho}}_s = -\frac{\varepsilon^2}{\nu_f} \sum_j \left([\sigma_j^- \sigma_{j+1}^+, [\sigma_j^+ \sigma_{j+1}^-, \hat{\bar{\rho}}_s]] + \text{h.c.}\right). \tag{23}$$

This is clearly a dissipative, Lindblad form, evolution coding for incoherent left / right hopping along the chain (which, as a model of incoherent hopping, could have been written directly without our journey through the stochastic XXZ model). It is independent of $\Delta$. See the Appendix F for details.

Equation (23) is a diffusive equation (it involves second order derivatives in the form of double commutators). The slow evolution of the local spins $S_j = \text{Tr}(\hat{\bar{\rho}}\sigma_j^z)$ reads $\partial_s \bar{S}_j = \text{Tr}(\sigma_j^z \mathfrak{A}\hat{\bar{\rho}})$. As shown in the Appendix F, it reduces to:

$$\partial_s \bar{S}_j = \text{Tr}\big(\sigma_j^z (\mathfrak{A}\hat{\bar{\rho}}_s)\big) = \frac{2\varepsilon^2}{\nu_f}(\bar{S}_{j+1} - 2\bar{S}_j + \bar{S}_{j-1}). \tag{24}$$

This is indeed a simple discrete diffusion equation (independent of the anisotropy parameter $\Delta$) with a diffusion constant inversely proportional to the friction coefficient, as expected from the classical considerations of Section 1.

## 3.3 The XXZ stochastic slow modes

Equation (23) describes the mean slow mode evolution. There are of course fluctuations, which we now describe. For any given realization of the Brownian motions, the evolution equation for the density matrix is

$$\rho_t \to \rho_{t+dt} = e^{-idH_t}\rho_t e^{+idH_t},$$

with $dH_t = h\,dt + \sqrt{\eta\nu_f}\sum_j \sigma_j^z dB_t^j$ with $h$ the XXZ hamiltonian. We may test this stochastic evolution by looking at the mean of any function $F(\rho_t)$ of the density matrix. For instance, we may consider polynomial functions, say $\text{Tr}(O_1\rho_t)\cdots\text{Tr}(O_p\rho_t)$, and look at their mean. This amounts to look for statistical correlations between operator expectations. Let $\mathbb{E}[F(\rho_t)]$ be their expectations (w.r.t. the the Brownian motions) of those functions. Their evolutions are coded in a Fokker-Planck like equation of the form

$$\partial_t \mathbb{E}[F(\rho_t)] = \mathbb{E}[\mathfrak{D}F(\rho_t)],$$

with $\mathfrak{D}$ a second order differential operator. It decomposes into $\mathfrak{D} = \eta\mathfrak{D}_1 + \mathfrak{D}_0$ where $\eta\mathfrak{D}_1$ is the Fokker-Planck operator associated to the noisy dynamics generated by the stochastic hamiltonian $\sqrt{\eta\nu_f}\sum_j \sigma_j^z dB_t^j$ and $\mathfrak{D}_0$ is the first order differential operator associated to the hamiltonian dynamics generated by the XXZ hamiltonian $h\,dt$. The explicit expression of those differential operators are given in Appendix E.

Let us now identify what the slow modes are. These modes are those whose expectations are non trivial in the large friction limit $\eta \to \infty$ at fixed slow time $s = t/\eta$. It is clear that the functions which are not in the kernel of $\mathfrak{D}_1$, i.e. those such that $\mathfrak{D}_1 F \neq 0$, have expectations which decrease exponentially fast in time $t$ with a time scale of order $\eta^{-1}$ – because their evolution equations are of the form $\partial_t \mathbb{E}[F(\rho_t)] = \eta\,\mathbb{E}[\mathfrak{D}_1 F(\rho_t)] + \cdots$ where $\cdots$ stand for sub-leading terms in $\eta^{-1}$. Functions which are annihilated by $\mathfrak{D}_1$ are those which are invariant under all local $U(1)$s generated by the $\sigma_j^z$'s. That is: $\text{Ker}\,\mathfrak{D}_1$ are made of $U(1)$s invariant functions. Let $\hat{\Pi}_0$ be the projector on $U(1)$s invariant functions. Perturbation theory then tells us that the induced dynamics on $\text{Ker}\,\mathfrak{D}_1$ is $\partial_t \mathbb{E}[F(\rho_t)] = \mathbb{E}[\hat{\Pi}_0\mathfrak{D}_0\hat{\Pi}_0 F(\rho_t)] + \cdots$ where the dots refer to sub-leading terms in $\eta^{-1}$. Hence, $U(1)$s invariant functions which are not in the kernel of $\hat{\Pi}_0\mathfrak{D}_0\hat{\Pi}_0$ also have a vanishing expectation in the limit $\eta \to \infty$ at fixed time $s = t/\eta$. Recall that $\mathfrak{D}_0$ is the differential operator associated to the hamiltonian dynamics generated by $h$. Since $h = h^{xy} + \Delta h^{zz}$ where $h^{zz}$ is $U(1)$s invariant but $h^{xy}$ is not, functions in $\text{Ker}\,\hat{\Pi}_0\mathfrak{D}_0\hat{\Pi}_0$ are the $U(1)$s invariant functions which are also invariant under the flow generated by $h^{zz}$.

In summary, the slow mode observables are the functions $F(\rho_t)$ of the density matrix which are invariant under all the local $U(1)$s generated by the $\sigma_j^z$'s and which are also invariant under the global $U(1)$ generated by $h^{zz}$. By construction, these functions are those invariant under conjugacy

$$F(\rho) = F(e^{-i\alpha h^{zz}-i\sum_j \theta_j \sigma_j^z}\rho\, e^{i\alpha h^{zz}+i\sum_j \theta_j \sigma_j^z}), \tag{25}$$

for any real parameters $\alpha$ and $\theta_j$'s. These functions are those which have non vanishing expectations in the large friction limit $\eta \to \infty$. For instance, products of local density expectations, say $\text{Tr}(n_{j_1}\rho_t)\cdots\text{Tr}(n_{j_p}\rho_t)$, are slow mode functions. But these are not the only the ones: more globally invariant functions can be constructed using the projectors $P^a_{j-1;j+2}$ defined in the following section (see below and Appendix E).

### 3.4 The effective stochastic slow dynamics of the stochastic XXZ model

Let us now determine the effective stochastic dynamics of the slow mode observables in the large friction limit (i.e. limit $\eta \to \infty$ at fixed $s = t/\eta$). Because the slow mode functions are made of functions invariant under conjugacy by the $\sigma^z_j$'s and $h^{zz}$, we can describe their dynamics using an interaction representation. Let us define $\hat{\rho}_t$ by

$$\hat{\rho}_t = e^{+iK_t}\rho_t e^{-iK_t}, \quad \text{with } K_t = t\Delta h^{zz} + \sqrt{\eta\nu_f}\sum_j \sigma^z_j B^j_t. \tag{26}$$

Going to this interaction representation is a way to absorb all the fast modes. By construction, if $F$ is a slow mode function then $F(\rho_t) = F(\hat{\rho}_t)$. So we can describe the time evolution of $F(\rho_t)$ by looking at that of $\hat{\rho}_t$.

The evolution equation for $\hat{\rho}_t$ is obtained from that of $\rho_t$ by conjugacy. Since the later is the stochastic unitary evolution generated by $dH_t = hdt + \sqrt{\eta\nu_f}\sum_j \sigma^z_j dB^j_t$ with $h = h^{xy} + \Delta h^{zz}$, we get

$$\hat{\rho}_{t+dt} = e^{-id\hat{H}_t}\hat{\rho}_t e^{+id\hat{H}_t}, \quad \text{with } d\hat{H}_t = e^{+iK_t}(h^{xy}dt)e^{-iK_t}, \tag{27}$$

where $K_t$ has been defined in eq.(26).

The aim of this Section is to describe what the hydrodynamic large friction limit is. As shown in Appendix C, it reduces to the stochastic unitary evolution, $\hat{\rho}_{s+ds} = e^{-id\hat{H}_s}\hat{\rho}_s e^{+id\hat{H}_s}$ with effective stochastic hamiltonian (w.r.t. the time $s = t/\eta$)

$$d\hat{H}_s = \sqrt{\frac{2\varepsilon^2}{\nu_f}}\sum_j \sum_{a=0,+,-} P^a_{j-1;j+2}(\sigma^+_j \sigma^-_{j+1})dW^{j;a}_s + \text{h.c.}, \tag{28}$$

where the $P^a_{j-1;j+2}$'s are projectors acting on sites $j-1$ and $j+2$ next to the link between sites $j$ and $j+1$. They are defined by

$$P^0_{k;l} = (\frac{1+\sigma^z_k}{2})(\frac{1+\sigma^z_l}{2}) + (\frac{1-\sigma^z_k}{2})(\frac{1-\sigma^z_l}{2}),$$
$$P^+_{k;l} = (\frac{1+\sigma^z_k}{2})(\frac{1-\sigma^z_l}{2}),$$
$$P^-_{k;l} = (\frac{1-\sigma^z_k}{2})(\frac{1+\sigma^z_l}{2}).$$

The $W^{j;a}_s$'s are complex Brownian motions normalized to

$$dW^{j;a}_s d\overline{W}^{k;b}_s = \delta^{j;k}\delta^{a;b} ds. \tag{29}$$

The evolution equation induced by the stochastic hamiltonian (28) can of course be written as the stochastic equation

$$d\hat{\rho}_s = -\frac{\varepsilon^2}{\nu_f}\sum_{j;a}\Big([\mathfrak{h}^a_j,[\mathfrak{h}^{a\dagger}_j,\hat{\rho}_s]] + \text{h.c.}\Big)ds - \sqrt{\frac{2\varepsilon^2}{\nu_f}}\sum_{j;a}\big(i[\mathfrak{h}^a_j,\hat{\rho}_s]dW^{j;a}_s + \text{h.c.}\big), \tag{30}$$

where the $\mathfrak{h}_j^a = P_{j-1;j+2}^a(\sigma_j^+\sigma_{j+1}^-)$ are the hopping operators from site $j+1$ to site $j$ dressed by the state values at neighbour sites.

The evolution equations for operators can be written similarly by duality. For an operator $O$, they read

$$dO_s = -\frac{\varepsilon^2}{\nu_f}\sum_{j;a}\Big(\big[\mathfrak{h}_j^a,[\mathfrak{h}_j^{a\dagger},O]\big]_s + \text{h.c.}\Big)ds + \sqrt{\frac{2\varepsilon^2}{\nu_f}}\sum_{j;a}\big(i\big[\mathfrak{h}_j^a,O\big]_s dW_s^{j;a} + \text{h.c.}\big). \qquad (31)$$

This equation can be simplified further if the operator $O$ commutes with all the $\sigma_j^z$'s. Indeed, then all projectors $P_{j-1;j+2}^a$ commute with $O$ and, since $\sum_a P_{j-1;j+2}^a = 1$, we can replace the drift term in eq.(30) by $-\frac{\varepsilon^2}{\nu_f}\sum_j\big(\big[\sigma_j^+\sigma_{j+1}^-,[\sigma_j^-\sigma_{j+1}^+,O]\big]_s + \text{h.c.}\big)ds$. In other words, for $O$ neutral w.r.t to the $U(1)$s actions generated by the $\sigma_j^z$'s, the drift term is $\Delta$-independent. For instance, for the local spin $\sigma_j^z$ we have

$$d\sigma_j^z(s) = \frac{2\varepsilon^2}{\nu_f}\big(\sigma_{j-1}^z(s) - 2\sigma_j^z(s) + \sigma_{j+1}^z(s)\big)ds + \sqrt{\frac{2\varepsilon^2}{\nu_f}}\big(d\mathbb{V}_s^j - d\mathbb{V}_s^{j-1}\big), \qquad (32)$$

with $d\mathbb{V}_s^j$ noisy operators of a specific form,

$$d\mathbb{V}_s^j = 2i\sum_{a=0,+,-}\big((\sigma_j^+\sigma_{j+1}^-)(s)\,dW_s^{j;a} - (\sigma_j^-\sigma_{j+1}^+)(s)\,d\overline{W}_s^{j;a}\big)P_{j-1;j+2}^a(s).$$

This has the appropriate –if not expected– structure: the drift term is the discrete diffusion operator (with constant diffusion coefficient) and the noise term is a discrete difference. Of course, the noise term drops out when looking at the mean evolution, and we recover eq.(24). The $\Delta$-dependence, reflected by the presence of the projectors $P_{j-1;j+2}^a$ – and hence the difference with the XY models – is manifest in higher moments of multipoint functions, i.e. in correlations of quantum expectations.

These evolution equations have a nice and simple interpretation: they code for incoherent hopping processes from one site to the next, either to the left or to the right. The probability to jump to the left or the right is dressed by the next nearest neighbour occupancies – there are the echoes of the operators $P_{j-1;j+2}^a$. Their impact can be thought of as introducing an effective, operator valued, diffusion constant, depending on the neighbour occupancies, and on the operators it is acting on. These dressings are absent in the XY model ($\Delta = 0$). Via the large friction limit, we have transmuted the on-site Brownian noise to Brownian processes attached to the links. Note that there is more than one Brownian motions (actually three in this case) attached to each link. As explained in the Introduction, this is a direct echo of the on-site randomness which destroys all phase coherences of the original XXZ hopping processes.

The proof of eqs.(28,30) is given in Appendix C.

## 3.5  An approximate classical fluctuating hydrodynamics

Here we present a –yet uncontrolled– approximation which reduces the quantum stochastic equation (32) to classical stochastic equations. The latter are of the form of fluctuating discrete hydrodynamic equations, similar to those consider in the macroscopic fluctuation theory.

Following the interpretation of the noisy interactions as encoding couplings between the spin chain and local baths attached to each of the lattice sites, it is natural to look for factorized approximations for the density matrix in the following form

$$\hat{\rho}_s =_{\text{anstaz}} \otimes_j\hat{\rho}_j(s), \quad \hat{\rho}_j(s) = \frac{1}{2}(1 + S_j(s)\sigma_j^z). \qquad (33)$$

This ansatz codes for some kind of local equilibration, breaking correlations between spins at distant sites. Of course such an ansatz is not (fully) compatible with the slow mode dynamics – in the sense that it is not preserved by eq.(30). It is only an approximation. In particular, it misses many correlations. As for any hydrodynamic approximation, it is expected to be valid if the typical correlation lengths or mean free paths are the shortest lengths of the problem. However, its domain of validity within the quantum stochastic model needs to be made more precise.

Within this approximation, the effective (classical) spins $S_j$ are the effective slow variables. The aim of this Section is thus to find an effective description of their slow dynamics which we encode into a stochastic differential equation. The drift term of this SDE is fixed by eq.(32). Thus we look for a stochastic equation of the form

$$dS_j = \frac{2\varepsilon^2}{\nu_f}\big(S_{j-1} - 2S_j + S_{j+1}\big)ds + \big(N_j d\tilde{V}_s^j - N_{j-1}d\tilde{V}_s^{j-1}\big), \tag{34}$$

with $\tilde{V}_s^j$ some effective Brownian motions, normalized by $d\tilde{V}_s^j d\tilde{V}_s^k = \delta^{j;k}ds$, and $N_j$ some $S$-dependent coefficients to be determined. Those coefficients cannot be directly determined by the dynamical equation (30) because the factorized ansatz (33) is not compatible with it – or alternatively, because $\text{Tr}(\hat{\rho}_j \sigma_j^{\pm}) = 0$ so that the noisy terms in eq.(32) disappear when averaging them against the factorized ansatz. Comparing the classical ansatz (34) and the quantum equation (32), it is natural to demand that the quadratic variations of the noise coincide, that is

$$\Big(\frac{2\varepsilon^2}{\nu_f}\Big)\big(d\mathbb{V}_s^j\big)^2 = N_j^2\, ds.$$

Actually, if this relation holds a more general one holds as well

$$\Big(\frac{2\varepsilon^2}{\nu_f}\Big)(d\mathbb{V}_s^j - d\mathbb{V}_s^{j-1})(d\mathbb{V}_s^k - d\mathbb{V}_s^{k-1}) = (N_j d\tilde{V}_s^j - N_{j-1}d\tilde{V}_s^{j-1})(N_k d\tilde{V}_s^k - N_{k-1}d\tilde{V}_s^{k-1}).$$

Imposing this rule specifies our approximation. This gives $N_j = 2\sqrt{\frac{\varepsilon^2}{\nu_f}}\sqrt{1 - S_j S_{j+1}}$. The approximate classical SDE thus reads

$$dS_j = \frac{2\varepsilon^2}{\nu_f}\big(S_{j-1} - 2S_j + S_{j+1}\big)ds + 2\sqrt{\frac{\varepsilon^2}{\nu_f}}\big(\sqrt{1 - S_j S_{j+1}}\,d\tilde{V}_s^j - \sqrt{1 - S_{j-1}S_j}\,d\tilde{V}_s^{j-1}\big). \tag{35}$$

This is a classical fluctuating discrete hydrodynamic equation. Its formal continuous limit can be taken without difficulty. The discrete (classical) variables $S^j$ are mapped to continuous variables $S_{(x,\tau)}$ with $x$ the space position ($x = aj$) and $\tau$ the hydrodynamic time ($\tau = a^2 s/\ell_0^2$), with $a$ the lattice mesh size and $\ell_0$ an arbitrary bare length scale. The drift term in eq.(35) clearly becomes a Laplacian and the noisy term a gradient. The hydrodynamic time $\tau = a^2 s/\ell_0^2$ is defined in such a way as to absorb the factor of $a$ arising through this mapping. One has to pay attention to the fact that the map of the discrete Brownian motions $d\tilde{V}^j$, with covariance $\delta^{i;j}ds$ to continuous Brownian white noise $d\zeta_{x,\tau}$, with covariance $d\zeta_{(x,\tau)}d\zeta_{(x',\tau)} = \delta(x - x')d\tau$ involves an extra factor $\sqrt{a}$ because $a^{-1}\delta^{i;j} \to \delta(x - x')$ in the continuous limit. As a consequence the naive continuous limit of eq.(35) w.r.t. to the hydrodynamic time $\tau$ is

$$dS_{(x,\tau)} = D_0\,\nabla_x^2 S_{(x,\tau)}\,d\tau + \sqrt{2aD_0}\,\nabla_x\big(\sqrt{1 - S_{(x,\tau)}^2}\,d\zeta_{(x,\tau)}\big), \tag{36}$$

with $D_0 = 2\varepsilon^2 \ell_0^2/\nu_f$ the effective diffusion constant. Notice the remaining (small) factor $\sqrt{a}$ weighting the white noise. Comparaison with the equation (1) for the classical macroscopic fluctuation theory is striking. However, in the quantum theory this equation is valid only within the classical hydrodynamic approximation.

## 3.6 The stochastic XY model

The quantum XY model is defined by the spin half Hamiltonian $h^{xy} = \sum_j h_j^{xy}$ with Hamiltonian density $h_j^{xy} = \varepsilon(\sigma_j^x \sigma_{j+1}^x + \sigma_j^y \sigma_{j+1}^y)$, that is

$$h^{xy} = \varepsilon \sum_j (\sigma_j^x \sigma_{j+1}^x + \sigma_j^y \sigma_{j+1}^y) = 2\varepsilon \sum_j (\sigma_j^+ \sigma_{j+1}^- + \sigma_j^- \sigma_{j+1}^+). \tag{37}$$

It is the Heisenberg Hamiltonian with zero anisotropy $\Delta = 0$. Thus, all results of the previous Section apply after setting $\Delta = 0$. As is well known, the XY model is equivalent to a free fermion model.

The stochastic XY model is defined by promoting the hamiltonian evolution to the stochastic unitary evolution generated by (recall that $2n_j = 1 + \sigma_j^z$):

$$dH_t = h^{xy} dt + 2\sqrt{\eta \nu_f} \sum_j n_j^z dB_t^j.$$

The evolution rules are $\rho_{t+dt} = e^{-idH_t} \rho_t e^{idH_t}$ for density matrices and $O_{t+dt} = e^{+idH_t} O_t e^{-idH_t}$ for operators. They can be written in a stochastic Lindblad form if needed. The mean evolution associated to these process has been studied in refs. [26] and [28] where a connection with Bethe ansatz was pointed out.

The previous discussion applies with $\Delta = 0$ up to a few points. First, the slow mode functions are simply functions invariant under all local $U(1)$s generated by the $n_j$'s. There is no extra projection on $h^{zz}$-invariant functions because the first order perturbation in $\eta^{-1}$ of the random dynamical flows is trivial, due to the structure of $h^{xy}$. Hence the only fast motion to absorb is the one generated by the local $U(1)$s. The interaction representation is then simply defined by $\hat{\rho}_t = e^{+i\tilde{K}_t} \rho_t e^{-i\tilde{K}_t}$, with $\tilde{K}_t = 2\sqrt{\eta \nu_f} \sum_j n_j^z B_t^j$. In the interaction picture, the time evolution reads

$$\hat{\rho}_{t+dt} = e^{-id\hat{H}_t} \hat{\rho}_t e^{+id\hat{H}_t},$$

with

$$d\hat{H}_t = e^{+i\tilde{K}_t}(h^{xy} dt) e^{-i\tilde{K}_t} = \sqrt{\frac{2\varepsilon^2}{\nu_f}} \sum_j d\tilde{W}_t^j(\eta) \sigma_j^+ \sigma_{j+1}^- + \text{h.c.}$$

with $d\tilde{W}_j(\eta) = \sqrt{2\nu_f}\, e^{2i\sqrt{\eta \nu_f}(B_t^j - B_t^{j+1})} dt$. Recall that $t = s\eta$. The main difference with the XXZ model is the absence of the conjugation by the hamiltonian $\Delta h^{zz}$. There is no phase proportional to the time $t$ which, in the XXZ model, comes from the conjugation with the hamiltonian $\Delta h^{zz}$. There is no extra decoherence induced by these phases associated to $\Delta h^{zz}$ and hence no extra projectors $P_{j-1;j+2}^a$. As explained in previous Section or in Appendix A, the noise $\tilde{W}_j(\eta)$ converges to normalized independent Brownian motions in the limit $\eta \to \infty$.

As a consequence, there is only one-Brownian motion per link in the large friction limit ($\eta \to \infty$ at $s = t/\eta$ fixed). And, in this limit, the effective stochastic XY hamiltonian reads (w.r.t to the slow time $s = t/\eta$)

$$d\hat{H}_s = \sqrt{\frac{2\varepsilon^2}{\nu_f}} \sum_j \left(\sigma_j^+ \sigma_{j+1}^- dW_s^j + \sigma_j^- \sigma_{j+1}^+ d\overline{W}_s^j\right),$$

with $W_s^j$ complex Brownian motions with Itô rules $dW_s^j d\overline{W}_s^k = ds$. The associated stochastic equation reads

$$d\hat{\rho}_s = -\frac{\varepsilon^2}{\nu_f} \sum_j \left([\sigma_j^- \sigma_{j+1}^+, [\sigma_j^+ \sigma_{j+1}^-, \hat{\rho}_s]] + \text{h.c.}\right) ds - \sqrt{\frac{2\varepsilon^2}{\nu_f}} \sum_j \left(i[\sigma_j^+ \sigma_{j+1}^-, \hat{\rho}_s] dW_s^j + \text{h.c.}\right).$$

$$\tag{38}$$

This is a simple and standard model of incoherent hopping. Again, the difference with the XXZ model resides the absence of the projectors $P^a_{j-1;j+2}$ dressing the hopping processes and the noise.

The evolution equations for the local spin $\sigma^z_j$ is given similarly:

$$d\sigma^z_j(s) = \frac{2\varepsilon^2}{\nu_f}\big(\sigma^z_{j-1}(s) - 2\sigma^z_j(s) + \sigma^z_{j+1}(s)\big)ds + \sqrt{\frac{2\varepsilon^2}{\nu_f}}\big(d\mathbb{V}^j_s - d\mathbb{V}^{j-1}_s\big), \tag{39}$$

with $d\mathbb{V}^j_s = 2i\big((\sigma^+_j\sigma^-_{j+1})dW^j - \text{h.c.}\big)$. This structure of this equation is simpler than that in the XXZ model in the sense it only involves the local current and the hamiltonian density. Again it possesses the appropriate, and expected, structure.

Note that the large friction limit and the $\Delta \to 0$ limit do not commute. This absence of commutativity reflects the fact that the XXZ dynamics induces more rapidly oscillating phases (related to the $\Delta h^{zz}$ eigenvalues) than the XY dynamics and hence it induces more decoherence.

## 4 Discussion and perspectives

We have described a framework to identify slow modes in dissipative quantum spin chains and their effective stochastic dynamics. Coupling a spin chain to quantum noise induces dissipative friction. In the limit of large friction the noise-induced dissipative processes project the states on a high dimensional manifold of slow modes. This mechanism is analogue to that used in reservoir engineering [24]. These slow modes are parametrized by local variables which we may view as quantum fields. The sub-leading asymptotic time evolution then generates a dissipative stochastic dynamics over the slow mode manifold, which can be described as an effective quantum dissipative (discrete) hydrodynamics.

Although we elaborated on the basic principles underlying the construction, we mainly concentrated on analysing the stochastic Heisenberg XXZ spin chain. Of course many questions remain to be studied –transport, finite size systems with or without boundary injection, boundary effects, robustness to perturbations, etc (see ref. [29]). We dealt with the effective theory at large friction –but studying the sub-leading contributions could also be interesting as they generate a non-linear diffusion constant [30]. In this limit the effective quantum stochastic dynamics that we identified are natural quantizations of the fluctuating discrete hydrodynamic equations. They could now be directly taken as starting points for modelling quantum diffusive transports and their fluctuations, but the detour we took through the large friction limit justified their precise structures –and part of them, say the dressing of the hopping operators in the case of the XXZ spin chain, would had been difficult to guess without this detour. It is interesting to notice that stochasticity within conformal field theory has recently been considered in [13].

To take the continuous limit of those discrete quantum hydrodynamic equations is of course an important step (see ref. [29]) –the continuous limit within the classical hydrodynamic approximation, making contact with the classical macroscopic fluctuation theory, is already quite under control. This will hopefully make contact with mesoscopic fluctuation theory, the quantum analogue of the macroscopic fluctuation theory, and will provide a way to question the statistical properties of a class of out-of-equilibrium quantum systems –transport fluctuations, their large deviation functions, etc.

We may also envision to extend the construction in continuous systems to develop a framework encompassing quantum hydrodynamics. Lattice sites would be replaced by the elementary cells over which hydrodynamic coarse graining is implemented. States factorization,

which is ultra-local in the Heisenberg spin chain, should be replaced by a factorization over the hydrodynamic cells. Requiring that the slow states are locally Gibbs-like would impose to choose the system-noise coupling operators to be proportional to the energy density, properly integrated over the hydrodynamic cells. We may also allow for current carrying states by diversifying the system-noise couplings.

**Note added**: While we were working on the material presented here, a related paper [31], developing parallel ideas but following different routes, was posted on arXiv. Since the approaches were different, we decided to present this note, dealing mainly with the effective stochastic quantum dynamics, and to postpone a more complete presentation of the mesoscopic fluctuation theory for a future paper [29].

## Acknowledgements

This work was in part supported by the ANR project "StoQ", contract number ANR-14-CE25-0003. D.B. thanks Herbert Spohn for discussions and for his interest in this work. We also thank Ohad Shpielberg for many discussions on this and related topics.

## A   Brownian transmutation

The section is devoted to the construction of Brownian motions from fast Brownian phases. We are going to prove that the random phases (16) converges to complex Brownian motions in the limit $\eta \to 0$.

To simplify the notation, and to be a bit more general, let $\vec{B}_t$ be a vector valued normalized real Brownian motion. That is, each of its component is a normalized Brownian motion and its different components are independent. We view $\vec{B}_t$ has valued in the Euclidean space, equipped with the Euclidean scalar product.

Let $\vec{a}$ be a real vector and $b$ a real scalar. We define

$$W_s^{\vec{a};b}(\eta) := \eta \int_0^s ds'\, e^{i\eta(\vec{a}\cdot\vec{B}_{s'}+bs')}, \tag{40}$$

or equivalently $dW_s^{\vec{a};b} := \eta\, ds\, e^{i\eta(\vec{a}\cdot\vec{B}_s+bs)}$. There are two contributions to the phases: the random phases $\eta\vec{a}\cdot\vec{B}_s$ and the deterministic phase $\eta bs$. They are both rapidly oscillating in the limit of large $\eta$. They interfere destructively in expectations unless the phases compensate exactly.

As a consequence, for any non vanishing vector $\vec{a}$, we have:
(i) The limits $W_s^{\vec{a};b} := \lim_{\eta\to\infty} W_s^{\vec{a};b}(\eta)$ exist.
(ii) The limiting processes are Brownian motions with covariances

$$\mathbb{E}[W_{s_1}^{\vec{a}_1;b_1} W_{s_2}^{\vec{a}_2;b_2}] = \Big(\frac{4}{\vec{a}_1^2}\Big)\, \delta^{\vec{a}_1+\vec{a}_2;\vec{0}}\, \delta^{b_1+b_2;0}\, \min(s_1,s_2), \tag{41}$$

or alternatively

$$dW_s^{\vec{a}_1;b_1}\, dW_s^{\vec{a}_2;b_2} = \Big(\frac{4}{\vec{a}_1^2}\Big)\, \delta^{\vec{a}_1+\vec{a}_2;\vec{0}}\, \delta^{b_1+b_2;0}\, ds. \tag{42}$$

The $W_s^{\vec{a};b}$'s are complex processes with $\overline{W_s^{\vec{a};b}} = W_s^{-\vec{a};-b}$.

Let us now explain the arguments for eqs.(41,42) – if not the proof. Our first aim is to show that the processes $s \to W_s^{\vec{a};b}$ are martingales (in the limit $\eta \to \infty$). There are several ways

to do it. The one we choose emphasizes how the $W_s^{\vec{a};b}$ (which for finite $\eta$ is a differentiable function) is close to a function with Brownian roughness. Notice that

$$M_s^{\vec{a};b}(\eta) := i \int_0^s \vec{a} \cdot d\vec{B}_{s'} \, e^{i\eta(\vec{a}\cdot\vec{B}_{s'}+bs')}, \tag{43}$$

as a stochastic Itô integral, is by construction a (complex) martingale with Brownian roughness. Apply Itô's formula to $e^{i\eta(\vec{a}\cdot\vec{B}_s+bs)}$ to get

$$e^{i\eta(\vec{a}\cdot\vec{B}_s+bs)} = \eta M_s^{\vec{a};b}(\eta) + (ib - \eta\vec{a}^2/2)W_s^{\vec{a};b}(\eta),$$

i.e.

$$W_s^{\vec{a};b}(\eta) = \frac{2}{\vec{a}^2}\frac{1}{1-\frac{2ib}{\eta\vec{a}^2}}M_s^{\vec{a};b}(\eta) - \frac{e^{i\eta(\vec{a}\cdot\vec{B}_s+bs)}}{\eta\vec{a}^2/2-ib}.$$

This means that the smooth $W_s^{\vec{a};b}(\eta)$ differs from the Brownian rough $\frac{2}{\vec{a}^2}M_s^{\vec{a};b}(\eta)$ by corrections of order $\eta^{-1}$ path-wise. In particular, to prove the claims on $W_s^{\vec{a};b}(\eta)$, it is enough to prove the corresponding claims on $M_s^{\vec{a};b}(\eta)$.

We take an arbitrary complex linear combination $M_s(\eta) := \sum_k \lambda_k M_s^{\vec{a}_k;b_k}(\eta)$ which is again a continuous martingale and set

$$\vec{U}_s(\eta) = \sum_k \lambda_k \vec{a}_k e^{i\eta(\vec{a}_k\cdot\vec{B}_s+b_k s)},$$

so that $dM_s(\eta) = \vec{U}_s(\eta) \cdot d\vec{B}_s$. Let $U$ be defined by $U = \sum_{k,l} \lambda_k \lambda_l (\vec{a}_k \cdot \vec{a}_l) \delta^{\vec{a}_k+\vec{a}_l;\vec{0}} \delta^{b_k+b_l;0}$.

The general theory of continuous martingales guaranties (via a direct application of Itô's formula) that $e^{M_s(\eta)-\frac{1}{2}\int_0^s ds' \vec{U}_{s'}^2(\eta)}$ is also a martingale, the exponential martingale of $M_s(\eta)$. The quadratic variation part

$$\int_0^s ds' \vec{U}_{s'}^2(\eta) = \sum_{k,l} \lambda_k \lambda_l (\vec{a}_k \cdot \vec{a}_l) \int_0^s ds' e^{i\eta((\vec{a}_k+\vec{a}_l)\cdot\vec{B}_{s'}+(b_k+b_l)s')}$$

is easily evaluated at large $\eta$. Take the $k,l$ term. Either $\vec{a}_k + \vec{a}_l = 0$ and $b_k + b_l = 0$ and then this term yields $\lambda_k \lambda_l (\vec{a}_k \cdot \vec{a})s$, independently of $\eta$, or the integrand is rapidly oscillating at large $\eta$ and the integral is small.[5] Thus

$$\lim_{\eta\to\infty} \int_0^s ds' \vec{U}_{s'}^2(\eta) = s \sum_{k,l} \lambda_k \lambda_l (\vec{a}_k \cdot \vec{a}_l) \delta^{\vec{a}_k+\vec{a}_l;\vec{0}} \delta^{b_k+b_l;0} = sU.$$

Hence $e^{M_s(\eta)-\frac{s}{2}U+o(\eta)}$ is a martingale.

Now it is an easy exercise in the manipulation of conditional expectations to prove that a process $X_s$ such that $e^{\lambda X_s-\lambda^2 s/2}$ is a martingale for every $\lambda$ has the finite dimensional distributions of a standard Brownian motion. Using the freedom of choice for the $\lambda_k$s, this implies that, at large $\eta$, the finite dimensional distributions of $M_s(\eta)$ are close to those of a rescaled Brownian motion. Recall that, at large $\eta$, $W_s^{\vec{a};b}(\eta) \sim \frac{2}{\vec{a}^2}M_s^{\vec{a};b}(\eta)$. Then, a glance at the formula for $U$ yields the normalizations in eqs.(41,42).

If one is not at ease with this formal manipulation, as a check one may compute the covariance of the $W$s to verify eq.(41).

One word of caution: we have used the Itô convention throughout, but one should keep in mind that if the smooth functions $W_s^{\vec{a};b}(\eta)$ are used as control/noise in differential equations, the large $\eta$ limit of these equations has to be interpreted in the Stratonovich convention. We took care of this fact in our computations.

---

[5]In fact, the integral is nothing but $\eta^{-1}W_s^{\vec{a}_k+\vec{a}_l;b_k+b_l}(\eta)$, so even if this look a bit like bootstrapping, it has to be small to be consistent with what we are proving, namely that the $W$s have finite limits at large $\eta$.

# B  A spin one-half toy model at strong noise

Here, we study a very simple toy model dealing with a spin one-half. The model is that of Rabi oscillations with random dephasing. By definition, the evolution equation for the density matrix is chosen to be

$$d\rho_t = -i\nu[\sigma^x, \rho_t]\,dt - \frac{\eta}{2}[\sigma^z,[\sigma^z,\rho_t]]\,dt - i\sqrt{\eta}\,[\sigma^z,\rho_t]\,dB_t,$$

with $B_t$ a normalized Brownian motion. This is a random unitary evolution, $\rho_t = U_t\rho_0 U_t^\dagger$, with unitaries $U_t$ generated by a random hamiltonian process $dH_t$,

$$U_{t+dt}U_t^\dagger = e^{-idH_t}, \quad dH_t = \nu\sigma^x\,dt + \sqrt{\eta}\,\sigma^z\,dB_t.$$

Let us parametrize the density matrix by $\rho = \frac{1}{2}(1 + \vec{S}\cdot\vec{\sigma})$ with $\vec{S}$ in the Bloch sphere (or more precisely Bloch ball): $\vec{S}^2 \leq 1$. The above equations are equivalent to

$$
\begin{aligned}
dS_t^z &= -2\nu S_t^y\,dt,\\
dS_t^x &= -2\eta S_t^x\,dt + 2\sqrt{\eta}\,S_t^y\,dB_t,\\
dS_t^y &= +2\nu S_t^z\,dt - 2\eta S_t^y\,dt - 2\sqrt{\eta}\,S_t^x\,dB_t.
\end{aligned}
$$

Because they code for random unitary transformations, these equations preserve the norm of the Bloch vector: $\vec{S}_t^2 = \text{constant}$.

Let us first look at the mean flow. Let $\bar{S}_t^a = \mathbb{E}[S_t^a]$. The evolution equations are simply obtained from those above by dropping the $dB_t$-terms. Hence, $d\bar{S}_t^x = -2\eta\bar{S}_t^x\,dt$ and

$$\bar{S}_t^x = \bar{S}_0^x\,e^{-2\eta t} \to 0,$$

as $\eta \to \infty$. The two other equations are coupled, $d\bar{S}_t^z = -2\nu\bar{S}_t^y\,dt$ and $d\bar{S}_t^y = +2\nu\bar{S}_t^z\,dt - 2\eta\bar{S}_t^y\,dt$. The solution is

$$2\nu\bar{S}_t^z + \lambda_\pm\bar{S}_t^y = e^{\lambda_\pm t}(2\nu\bar{S}_0^z + \lambda_\pm\bar{S}_0^y),$$

with $\lambda_\pm = -\eta \pm \sqrt{\eta^2 - 4\nu^2}$ the two eigen-values of the linear problem. For large $\eta$, we have $\lambda_- \simeq -2\eta$ and $\lambda_+ \simeq -2\nu^2/\eta$. From this we see that

$$\bar{S}_t^y \simeq \bar{S}_0^y\,e^{-2\eta t} \to 0, \quad \bar{S}_t^z \simeq \bar{S}_0^z\,e^{-2\nu^2(t/\eta)},$$

asymptotically in $\eta$. That is: only the component along the $z$-axis survives in the large $\eta$ limit with a non trivial dynamics w.r.t. the time $s = t/\eta$. This is the mean slow mode dynamics.

Let us now look at higher moments, or more generally at the expectation of any function $F(\vec{S}_t)$. As is well know, the time evolution of those expectations is governed by a (dual) Fokker-Planck operator $\mathfrak{D}$ via $\partial_t\mathbb{E}[F(\vec{S}_t)] = \mathbb{E}[(\mathfrak{D}F)(\vec{S}_t)]$. In the present case, this operator reads

$$\mathfrak{D} = -2\nu i\mathscr{D}_x - 2\eta\mathscr{D}_z^2,$$

with $\mathscr{D}_x = i(S^z\partial_{S^y} - S^y\partial_{S^z})$ and $\mathscr{D}_z = i(S^y\partial_{S^x} - S^x\partial_{S^y})$ the differential operators generating rotations around the $x$- and $z$-axis respectively. The spectrum of $\mathscr{D}_z^2$ can be easily found, say by decomposing $F$ on spherical harmonics. It is positive (made of non-negative integers). Thus the only functions whose expectation does not vanish in the limit $\eta \to \infty$ are those annihilated by $\mathscr{D}_z$. The others have exponentially small expectations.

Hence, the slow mode observables are the functions $F(\vec{S}_t)$ invariant by rotation around the $z$-axis. These are functions of the two fundamental invariants $S^z$ and $\sqrt{\vec{S}^2}$. Via a rotation

around the $z$-axis, any point in the Bloch sphere can be mapped onto a point in the half disc say $\mathbb{D} = \{S^y = 0, S^x \geq 0, \vec{S}^2 \leq 1\}$ – or any other equivalent half disc obtained from that one by rotation around the $z$-axis. Alternatively, any orbit of the rotation group around the $z$-axis in the Bloch sphere intersects $\mathbb{D}$ once, and only once. Points on this half disc thus parametrize these orbits and $S^z$ and $\sqrt{\vec{S}^2}$ are local coordinates on $\mathbb{D}$.

The slow mode process is that of $S^z$ and $\sqrt{\vec{S}^2}$ in the limit $\eta \to \infty$, w.r.t. to the time $s = t/\eta$. It takes place on the half disc $\mathbb{D}$. To find it we go to the interaction representation which amounts to conjugate all quantum observables by the random $z$-rotation $e^{i\sqrt{\eta}\sigma^z B_t}$:

$$\hat{\rho}_t = e^{i\sqrt{\eta}\sigma^z B_t} \rho_t \, e^{-i\sqrt{\eta}\sigma^z B_t}.$$

Let $\hat{\rho}_t = \frac{1}{2}(1 + \hat{\vec{S}}_t \cdot \vec{\sigma})$. Of course $\hat{S}_t^z = S_t^z$ and $\hat{\vec{S}}_t^2 = \hat{\vec{S}}_t^2$. In this transformed frame, the evolution is still unitary with random hamiltonian

$$d\hat{H} = e^{i\sqrt{\eta}\sigma^z B_t} (\nu\sigma^x \, dt) e^{-i\sqrt{\eta}\sigma^z B_t}$$
$$= \nu\big(\sigma^+ \, dW_s(\eta) + \sigma^- \, d\overline{W}_s(\eta)\big),$$

with $dW_s(\eta) = e^{i2\sqrt{\eta}B_t} dt$ and $d\overline{W}_s(\eta)$ its complex conjugate. Since $\sqrt{\eta}B_t = \eta B_s$ in law, with $s = t/\eta$, we may alternatively write $dW_s(\eta) = e^{i2\eta B_s} \eta ds$. As proved in Appendix A, these processes converge to complex Brownian motions $dW_s$ with $dW_s d\overline{W}_s = ds$. In the interaction representation, the evolution equation in the large $\eta$ limit is thus (by Itô calculus)

$$d\hat{\rho}_s = -i[d\hat{H}_s, \hat{\rho}_s] - \frac{1}{2}[d\hat{H}_s, [d\hat{H}_s, \hat{\rho}_s]].$$

For the two gauge invariant coordinates $S^z$ and $\vec{S}^2$, this yields,

$$d\vec{S}^2 = 0,$$
$$dS_s^z = -2\nu^2 S_s^z \, ds - i\nu(\hat{S}_s^+ dW_s - \hat{S}_s^- d\overline{W}_s),$$

with $\hat{S}^\pm = \hat{S}^x \pm i\hat{S}^y$. We may then follow two different routes. Either we fix the gauge, say $\hat{S}_s^y = 0$, so that $\hat{S}_s^+ = \hat{S}_s^- = \sqrt{\vec{S}_s^2 - (S_s^z)^2}$, and notice that $d\tilde{B}_s = i(dW_s - d\overline{W}_s)/\sqrt{2}$ is a normalized Brownian motion. Or, we observe that $i(\hat{S}_s^+ dW_s - \hat{S}_s^- d\overline{W}_s)$ is proportional to a Brownian increment (it is a martingale): $i(\hat{S}_s^+ dW_s - \hat{S}_s^- d\overline{W}_s) = \sqrt{2(\vec{S}_s^2 - (S_s^z)^2)} \, d\tilde{B}_s$ in law. Both routes yield the same gauge invariant equations:

$$d\vec{S}_s^2 = 0,$$
$$dS_s^z = -2\nu^2 S_s^z \, ds - \nu\sqrt{2\big(\vec{S}^2 - (S_s^z)^2\big)} \, d\tilde{B}_s,$$

with $\tilde{B}_s$ a real Brownian motion with $d\tilde{B}_s^2 = ds$. The mean flow is of course identical to that we found above. Let us stress again that this is a gauge invariant form of a process on the space of the orbits (of the group of $z$-rotations) in the Bloch sphere parametrized by points in the half disc $\mathbb{D}$.

## C  Proof of the XXZ stochastic slow mode dynamics

Let us now argue for eqs.(28,30). We start from eq.(27) which codes for the dynamics in the interaction representation at finite friction $\eta$. By decomposing $K_t$ as $K_t = t\Delta h^{zz} + \sqrt{\eta\nu_f}\sum_j \sigma_j^z B_t^j$, this can be written as (recall that $s = t/\eta$)

$$d\hat{H}_s = \sqrt{\frac{2\varepsilon^2}{\nu_f}} \sum_j e^{iK_s^{zz}} (\sigma_j^+ \sigma_{j+1}^-) e^{-iK_s^{zz}} \, d\tilde{W}_s^j(\eta) + \text{h.c.}$$

with $K_t^{zz} = \eta s \Delta h^{zz}$ and

$$d\tilde{W}_j(\eta) = \sqrt{2\nu_f}\, e^{2i\sqrt{\eta\nu_f}(B_{\eta s}^j - B_{\eta s}^{j+1})}\, \eta ds.$$

The adjoint action of $K_s^{zz}$ on $\sigma_j^+ \sigma_{j+1}^-$ can be computed exactly using the commutation relation $[h^{zz}, \sigma_j^+ \sigma_{j+1}^-] = 2(\sigma_{j-1}^z - \sigma_{j+2}^z)(\sigma_j^+ \sigma_{j+1}^-)$. We then get the alternative expression

$$d\hat{H}_s = \sqrt{\frac{2\varepsilon^2}{\nu_f}} \sum_j (\sigma_j^+ \sigma_{j+1}^-)\, d\mathbb{W}_s^j(\eta) + \text{h.c.},$$

where $d\mathbb{W}_j(\eta)$ is an operator valued process defined, at finite $\eta$, by

$$d\mathbb{W}_s^j(\eta) = e^{i2\eta\Delta\varepsilon(\sigma_{j-1}^z - \sigma_{j+2}^z)s}\, d\tilde{W}_s^j(\eta).$$

The projectors $P_{j-1;j+2}^a$ are the projectors on the eigen-spaces of $(\sigma_{j-1}^z - \sigma_{j+2}^z)$ with eigenvalues $2a$ for $a = 0, +, -$. Thus

$$d\mathbb{W}_s^j(\eta) = \sum_{a=0,+,-} P_{j-1;j+2}^a\, d\hat{W}_s^{j;a} \quad \text{with } d\hat{W}_s^{j;a} = e^{i4a\eta\Delta\varepsilon s}\, d\tilde{W}_s^j(\eta).$$

Using that $B_{\eta s}^j = \sqrt{\eta}\, B_s^j$ in law, we can write $d\hat{W}_s^{j;a}$ as

$$d\hat{W}_s^{j;a} = \sqrt{2\nu_f}\, e^{i\eta\left(2\sqrt{\nu_f}(B_s^j - B_s^{j+1}) + i4a\Delta\varepsilon s\right)}\, \eta ds.$$

Now, we recognized in this formula the fast Brownian phases that we studied in Appendix A. There we proved that they converge to complex Brownian motion. This ends the proof of eqs.(28,30).

## D  Derivation of the slow mode mean dynamics

Here, we present the derivation of the effective equation (9) for the mean slow modes. Recall that $d\bar{\rho}_t = (L + \eta L_b)(\bar{\rho}_t))dt$ with $L(\rho) = -i[h, \rho] + \eta L_s(\rho)$. The Lindbladian $L_b$ is negative (as a Lindbladian should be). Let $\Pi_0$ the projector on $\text{Ker}L_b$. Any density matrix $\rho$ may be decomposed into its component on $\text{Ker}L_b$ and its (orthogonal) complement: $\rho = \rho^{\parallel} + \rho^{\perp}$ with $\rho^{\parallel} = \Pi_0 \rho \in \text{Ker}L_b$ and $\rho^{\perp} = (1 - \Pi_0)\rho$. Since $\lim_{\eta\to\infty} e^{t\eta L_b} = \Pi_0$, we look for an expansion of the density matrix in the form $\bar{\rho} = \bar{\rho}_0 + \eta^{-1}\bar{\rho}_1 + \eta^{-2}\bar{\rho}_2 + \cdots$ with $\bar{\rho}_0 \in \text{Ker}L_b$. Writing the evolution equation, $\partial_t \bar{\rho}_t = (\eta L_b + L)(\bar{\rho}_t)$ order by order in $\eta^{-1}$ yields:

$$L_b(\bar{\rho}_0) = 0,$$
$$\partial_t \bar{\rho}_0 = L(\bar{\rho}_0) + L_b(\bar{\rho}_1),$$
$$\partial_t \bar{\rho}_1 = L(\bar{\rho}_1) + L_b(\bar{\rho}_2), \quad \text{etc}\ldots$$

The first equation says that $\bar{\rho}_0 \in \text{Ker}L_b$. Projecting the second equation on $\Pi_0$ gives $\partial_t \bar{\rho}_0 = \Pi_0 L(\bar{\rho}_0)$. Projecting it on the complement of $\text{Ker}L_b$ determines $\bar{\rho}_1$ up to its component in $\text{Ker}L_b$ which remains undetermined:

$$\bar{\rho}_1 = \bar{\rho}_1^{\parallel} + \bar{\rho}_1^{\perp}, \quad (L_b \bar{\rho}_1)^{\perp} = -(L \bar{\rho}_0)^{\perp},$$

with $\bar{\rho}_1^{\parallel} \in \text{Ker}L_b$. Projecting the last equation on $\Pi_0$ gives $\partial_t \bar{\rho}_1^{\parallel} = \Pi_0 L(\bar{\rho}_1)$.

Let us now assume that $\Pi_0 L \Pi_0 = 0$, as otherwise we would have to redefine the slow modes to take into account the dynamical flow it generates. Since by construction $L_b$ is invertible on

– and onto – the complement of $\mathrm{Ker}L_b$, the relation $(L_b\bar{\rho}_1)^\perp = -(L\bar{\rho}_0)^\perp$ can alternatively be written as $\bar{\rho}_1^\perp = -(L_b^\perp)^{-1}L\Pi_0\bar{\rho}_0$. Then $\partial_t\bar{\rho}_0 = 0$ and

$$\partial_t\bar{\rho}_1^\parallel = \Pi_0 L(\bar{\rho}_1) = -(\Pi_0 L\,(L_b^\perp)^{-1}\,L\Pi_0)(\bar{\rho}_0).$$

To leading order in $\eta^{-1}$, this is equivalent to (recall that $s = t/\eta$)

$$\partial_s\bar{\rho}_t = \eta\partial_t\bar{\rho}_t = \mathfrak{A}\bar{\rho}_t,$$

with $\mathfrak{A}\rho = -(\Pi_0 L\,(L_b^\perp)^{-1}\,L\Pi_0)(\rho)$. This proves eq.(9).

Alternatively, and to make the previous computation more concrete, let us assume –this is the case in all examples we discussed– that $L_b$ is diagonalizable. Let $L_b = \sum_{\nu\leq 0}\nu\Pi_\nu$, with $\Pi_\nu\Pi_{\nu'} = \delta_{\nu,\nu'}\Pi_\nu$, be its spectral decomposition. Let $\rho = \sum_\nu\rho^{(\nu)}$ be the decomposition of a density matrix $\rho$ onto its $L_b$-eigen components, $\rho^{(\nu)} = \Pi_\nu\rho$. Then $\rho^\parallel = \rho^{(0)}$. The relation between $\bar{\rho}_0$ and $\bar{\rho}_1$ then reads $\bar{\rho}_1 = \bar{\rho}_1^{(0)} - \sum_{\nu\neq 0}\frac{1}{\nu}(\Pi_\nu L\Pi_0)(\bar{\rho}_0)$. The inverse of $L_b$ on the complement of $\mathrm{Ker}L_b$ is then defined by $(L_b^\perp)^{-1} = \sum_{\nu\neq 0}\nu^{-1}\Pi_\nu$. The evolution equation for $\bar{\rho}_1$ then reads

$$\partial_t\bar{\rho}_1 = -\sum_{\nu\neq 0}\frac{1}{\nu}(\Pi_0 L\Pi_\nu L\Pi_0)(\bar{\rho}_0) = -(\Pi_0 L\,(L_b^\perp)^{-1}\,L\Pi_0)(\bar{\rho}_0).$$

To leading order in $\eta^{-1}$ this is equivalent to $\partial_t\bar{\rho}_t = \eta^{-1}\mathfrak{A}\bar{\rho}_t$, with $\mathfrak{A}\rho = -\sum_{\nu\neq 0}\frac{1}{\nu}(\Pi_0 L\Pi_\nu L\Pi_0)(\rho)$ as above.

# E Strong noise limit and effective stochastic dynamics

Here we discuss the large friction limit of stochastic dynamics of the stochastic spin chains and describe how to determine the slow mode observables and their effective dynamics via a second order perturbation theory on Fokker-Planck operators.

Let us first introduce simple differential operators acting on functions $F(\rho)$ of the density matrix. For any operator $X$ acting on the system Hilbert space, let $\mathscr{D}_X$ be the differential operator acting on functions $F(\rho)$ via

$$(\mathscr{D}_X F)(\rho) = \frac{d}{du}F(e^{-uX}\rho e^{uX})|_{u=0}.$$

For instance, if $F$ is a linear function, say $F(\rho) = \mathrm{Tr}(O\rho)$, then $(\mathscr{D}_X F)(\rho) = -\mathrm{Tr}(O[X,\rho]) = \mathrm{Tr}([X,O]\rho)$. If $X$ and $Y$ are two operators then $[\mathscr{D}_X,\mathscr{D}_Y] = \mathscr{D}_{[X,Y]}$.

Let us consider the stochastic differential equation (5) which we recall here for simplicity in the special case $L_s = 0$ (the generalisation to $L_s \neq 0$ is simple):

$$d\rho_t = -i[h,\rho_t]\,dt + \eta\,L_b(\rho_t)\,dt + \sqrt{\eta}\sum_j D_j(\rho_t)\,dB_t^j,$$

with $D_j(\rho) = -i[e_j,\rho]$ and $L_b(\rho) = -\frac{1}{2}\sum_j[e_j,[e_j,\rho]]$. Let $F$ be any (regular enough) function over density matrices. A simple application of Itô calculus yields that

$$dF(\rho_t) = (\mathfrak{D}F)(\rho_t)\,dt + i\sqrt{\eta}\sum_j(\mathscr{D}_{e_j}F)(\rho_t)\,dB_t^j,$$

with $\mathfrak{D}$ the second order differential operator, dual of the Fokker-Planck operator, defined by

$$\mathfrak{D} = i\mathscr{D}_h - \frac{\eta}{2}\sum_j\mathscr{D}_{e_j}^2. \tag{44}$$

We set $\mathfrak{D} = \eta\,\mathfrak{D}_1 + \mathfrak{D}_0$ with $\mathfrak{D}_0 = i\mathscr{D}_h$ and $\mathfrak{D}_1 = -\frac{1}{2}\sum_j \mathscr{D}_{e_j}^2$. Both $\mathfrak{D}$ and $\mathfrak{D}_1$ are negative operators.

Notice that the hydrodynamics limit of large friction is a limit of strong noise.

As is well known, the operator $\mathfrak{D}$ governs the time evolution of expectations: $\partial_t \mathbb{E}[F(\rho_t)] = \mathbb{E}[(\mathfrak{D}F)(\rho_t]$. Given the initial value $\rho_0$, its formal solution is: $\mathbb{E}[F(\rho_t)] = \left(e^{t(\eta\mathfrak{D}_1 + \mathfrak{D}_0)}\right)F(\rho_0)$. Hence, the only functions whose expectation survives in the limit $\eta \to \infty$ (at fixed $s = t/\eta$) are those in the kernel of $\mathfrak{D}_1$ if $\hat{\Pi}_0\mathfrak{D}_0\hat{\Pi}_0 = 0$ with $\hat{\Pi}_0$ the projector in $\mathrm{Ker}\mathfrak{D}_1$. If $\hat{\Pi}_0\mathfrak{D}_0\hat{\Pi}_0 \neq 0$, the function has to be both in $\mathrm{Ker}\mathfrak{D}_1$ and in $\mathrm{Ker}\,\hat{\Pi}_0\mathfrak{D}_0\hat{\Pi}_0$ in order to have non trivial expectation in the hydrodynamic limit. These are the slow mode observables.

The effective evolution (w.r.t. the time $s = t/\eta$) of the slow mode observables can then be derived via a perturbation expansion to second order in $\eta^{-1}$ parallel to that done in the previous Appendix D but dealing with the operators $\mathfrak{D}_0$ and $\mathfrak{D}_1$ acting on functions instead of the Lindbladian operators. This yields eq.(12).

Let us finish this Appendix by giving a few examples of slow mode observables in the case of the XXZ model. Of course there are all the functions of the local spins $\sigma_j^z$'s – or alternative the local densities $n_j = \frac{1}{2}(1 + \sigma_j^z)$:

$$\mathrm{Tr}(\rho_t\,\sigma_{j_1}^z \cdots \sigma_{j_p}^z)\cdots\mathrm{Tr}(\rho_t\,\sigma_{k_1}^z \cdots \sigma_{k_q}^z),$$

and their multi-time analogues. But one may also consider products of expectations of the local lowering / raising spin operators $\sigma_j^{\pm}$'s. Neutrality with respect to all the $U(1)$s generated the $\sigma_j^z$'s is easy to ensure. Since $[h^{zz}, \sigma_j^{\pm}] = \pm 2\varepsilon(\sigma_{j-1}^z + \sigma_{j+1}^z)\sigma_j^{\pm}$, let us introduce the projectors $Q_{j-1,j+1}^a$ on the eigen-space of $(\sigma_{j-1}^z + \sigma_{j+1}^z)$ with eigen-value $2a$, so that $[h^{zz}, Q_{j-1;j+1}^a \sigma_j^{\pm}] = \pm 4a\,Q_{j-1;j+1}^a\sigma_j^{\pm}$. Let $\Sigma_j^{a;\epsilon} = Q_{j-1;j+1}^a \sigma_j^{\epsilon}$. Then, products of expectations of those operators

$$\mathrm{Tr}(\rho_t\,\Sigma_{j_1}^{a_{j_1};\epsilon_{j_1}} \cdots \Sigma_{j_p}^{a_{j_p};\epsilon_{j_p}})\cdots\mathrm{Tr}(\rho_t\,\Sigma_{k_1}^{a_{k_1};\epsilon_{k_1}} \cdots \Sigma_{k_p}^{a_{k_p};\epsilon_{k_p}})$$

are slow mode observables provided there is global neutrality with respect to all the $U(1)$s generated by the $\sigma_j^z$'s and $\sum_{j_n} a_{j_n}\epsilon_{j_n} + \cdots + \sum_{k_n} a_{k_n}\epsilon_{k_n} = 0$. For instance, $\mathrm{Tr}(\rho_t\,\Sigma_j^{0;+})\mathrm{Tr}(\rho_t\,\Sigma_j^{0;-})$. Similarly one may construct slow modes observables using the local densities $(P_{j-1;j+2}^a\sigma_j^+\sigma_{j+1}^-)$ introduce in the text.

## F  Derivation of the XXZ mean diffusive equation

We give here details concerning the computation of the mean diffusive equation for the stochastic Heisenberg model, eqs.(22,23). In this case, the dissipative Lindbladian $L_b$ is a sum of local terms, $L_b = \sum_j L_b^j$ with $L_b^j(\rho) = -\frac{\nu_f}{2}[\sigma_j^z, [\sigma_j^z, \rho]]$. All $L_b^j$'s commute, $[L_b^j, L_b^k] = 0$. The spectrum of the Lindbladian $L_{\mathrm{loc}}(\rho) = -\frac{\nu_f}{2}[\sigma^z, [\sigma^z, \rho]]$ is made of 0 and $-2\nu_f$. Both eigenvalues are twice degenerate with $1, \sigma^z$ with eigenvalue 0 and $\sigma^x, \sigma^y$ with eigenvalue $-2\nu_f$. The eigenvalues of $L_b$ are thus $-2k\nu_f$ with $k = 0, \cdots, N$, with $N$ the number of sites, and $L_b$ acts diagonally on the operator basis $\sigma_1^{a_1}\sigma_2^{a_2}\cdots\sigma_N^{a_N}$ (with the convention $\sigma^0 = 1$). It is thus simple to compute the action of $L_b$ and of $(L_b^\perp)^{-1}$.

Let us first argue that states in $\mathrm{Ker}L_b$ are the density matrices with local components diagonal in the $\sigma_j^z$'s basis. Indeed, since $L_b$ is a sum of commuting operators, $L_b = \sum_j L_b^j$, where each $L_b^j$ is a negative operator, we have $\mathrm{Ker}\,L_b = \cap_j\mathrm{Ker}L_b^j$. Particular invariant states are factorized states of the form $\otimes_j \frac{1}{2}(1 + S_j\sigma_j^z)$.

Let us now evaluate $\mathfrak{A}\bar{\rho}$ with $\bar{\rho} \in \mathrm{Ker}L_b$. Recall that $\mathfrak{A} = -(\Pi_0 L (L_b^{\perp})^{-1} L \Pi_0)$ with $\Pi_0$ the projector on $\mathrm{Ker}L_b$. Recall that $L(\rho) = -i[h, \rho]$ with $h = h^{xy} + \Delta h^{zz}$. For $\bar{\rho} \in \mathrm{Ker}L_b$ we have $[h, \bar{\rho}] = [h^{xy}, \bar{\rho}]$ since the contribution from $\Delta h^{zz}$ vanishes. Then notice that, for $\bar{\rho} \in \mathrm{Ker}L_b$, we have $\Pi_0 L(\bar{\rho}) = 0$ and that $L(\bar{\rho})$ is an eigenvector of $L_b$ with eigenvalue $-4\nu_f$. Hence $(L_b^{\perp})^{-1} L(\bar{\rho}) = -(4\nu_f)^{-1} L(\bar{\rho})$. Thus, for $\bar{\rho} \in \mathrm{Ker}L_b$:

$$\mathfrak{A}\bar{\rho} = -(4\nu_f)^{-1} \Pi_0[h, [h, \bar{\rho}]],$$

as claimed in eq.(22).

Let us now compute this double commutator. By evaluating the $U(1)$s charges of the double commutator, it is clear that $\Pi_0[h^{zz}, [h, \bar{\rho}] = 0$ for any $\bar{\rho} \in \mathrm{Ker}L_b$. Hence $\Pi_0[h, [h, \bar{\rho}]] = \Pi_0[h^{xy}, [h^{xy}, \bar{\rho}]]$ and it is $\Delta$-independent. Let us decompose $h^{xy}$ as $h^{xy} = \sum_j h_j^{xy} = 2\varepsilon \sum_j (\sigma_j^+ \sigma_{j+1}^- + \sigma_j^- \sigma_{j+1}^+)$. Then again by evaluating the $U(1)$s charges of the double commutator and keeping only the terms with zero $U(1)$s charges, it is clear that the double commutator $\Pi_0[h^{xy}, [h^{xy}, \bar{\rho}]]$ reduces to

$$\Pi_0[h^{xy}, [h^{xy}, \bar{\rho}]] = 4\varepsilon^2 \sum_j \big([\sigma_j^+ \sigma_{j+1}^-, [\sigma_j^- \sigma_{j+1}^+, \bar{\rho}]] + \mathrm{h.c.}\big),$$

for any $\bar{\rho} \in \mathrm{Ker}L_b$. This proves eq.(23). The proof of eq.(24) is then direct, using $S_j = \mathrm{Tr}(\sigma_j^z \bar{\rho})$.

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
