# Peer review of "Stochastic dissipative quantum spin chains (I) : Quantum fluctuating discrete hydrodynamics"

_SciPost Physics, doi:SciPost Phys. 3, 033 (2017)_

## Round 2 · Referee Report · Anonymous (Referee 1) · 2017-8-10

Strengths

- Interesting and timely topic
- Novel aspects on dissipative dynamics of XXZ chain

Weaknesses

- Introduction poorly written and hard to understand
- Better comparison to standard Lindblad formalism should be provided

Report

The authors consider stochastic dynamics of quantum chains,
using the methods of quantum stochastic calculus.
In particular, they are interested in the limit of large friction,
i.e. when the coefficient of the dissipative coupling to an environment
is much larger in magnitude then the couplings generating the
coherent part of the time evolution. After introducing the
general formalism of stochastic calculus, they study the slow-modes
of the dynamics in a perturbative expansion w.r.t. the inverse friction
coefficient. The concrete example of an XXZ chain with a dephasing
noise is considered afterwards, where the slow-mode observables
are explicitly constructed and the evolution equations for the
density matrix and observables are presented.

In my opinion, although the paper is rather technical and thus somewhat
difficult to read, it contains some interesting novel results which
are worth publishing. There are, however, certain amendments that
should be considered in order to improve the quality of the
manuscript.

Requested changes

1)
The introduction is not well written. In particular, I do not
understand why the various evolution equations are presented
here. On one hand, these are impossible to comprehend without
any clue about the notation and the underlying methods.
On the other hand, this only makes the manuscript repetitive,
the introduction lengthy, without providing a better insight
to the reader. Some more detailed general introduction to
quantum stochastic differential equations (e.g. what are the
main assumptions to their applicability? where are they used?)
would be much more instructive at this point than quoting the
results for the XXZ chain. I would suggest the authors to
extensively revise their introductory chapter.

2)
I found the notion of “stochastic Lindblad equation” rather confusing.
The Lindblad master equation describes the evolution of a system
density matrix interacting with its environment in a Markovian
approximation, after integrating out the bath degrees of freedom.
The Lindblad dynamics describes coherent as well as stochastic
processes, resulting in a $non$-$unitary$ time-evolution of the
density matrix. In contrast, the stochastic differential equations
the authors consider describe a $unitary$ time evolution of the
density matrix. After an averaging over the noise terms they
indeed yield the Lindblad equation. However, this difference should
be made clear from the very beginning, or one should rather stick to
“stochastic differential equation” when referring to the noisy
dynamics.

3)
It did not become completely clear to me what the advantage of working
with the stochastic differential equation formalism really is?
If I understood it correctly, the Lindblad generators completely fix
the form of the noise term. On the other hand, when considering
expectation values, the noise has to be averaged over anyway.
What extra information can one expect to have on the dynamics of the
density operator and physical observables which is not there when
working with a Lindblad master equation? I believe this issue
deserves some more clarification.

4)
The word “hopping” is misspelled throughout the text.

---

## Round 3 · Author Response

1- As suggested by the referee, we modified the introduction, which is now shorter and hopefully clearer and less technical. To satisfy the referee's demand, we have suppressed some of the equations present in the introduction. But we believe that there is some usefulness in presenting the structure of the equations governing the phenomena we aim at describing, even-though some of the terms entering those equations are loosely defined at this stage (they are fully defined in the text).

2- The term « stochastic Lindblad equation » was made in analogy with the already existing name « stochastic Schroedinger equation » which codes for some stochastic deformation of the Schrodinger equation. The equations we deal with are stochastic extensions of the Lindblad equation (which is a deterministic, non-random, evolution equation). There is not a unique way to consistently extend stochastically a Lindblad equation given its deterministic part. For instance, quantum trajectory equations and our equations are both stochastic equations and have identical deterministic drift parts. Thus we had to introduce a new name to differentiate both types of equations which describe different physical processes. Thanks to the referee comment, we understand that the name we choose did not reach our aim, so we remove it except at one instance.

3- We are surprised by this comment (for instance, one would not ask this question in the case of classical stochastic equations of the Langevin type). We are dealing with the stochastic differential equation because it codes for the fluctuations of all observable quantities. The Lindblad equation, which is the deterministic part of the equation, only codes for the evolution of the mean (over the noise) of those quantities (in quantum theory there are two origins of randomness: that due to the noise and that due to the probablistic nature of quantum mechanics). This is standard in probability theory and does not require further explanation. Let us also point out that it is in general not true that the Lindblad generator completely fixes the form of the noise term of the density matrix evolution equation: to fix this term given the drift term one has to invoque extra information, say specifying which physical processes are at play (e.g. monitoring or unitary interaction with extra degrees of freedom), or specifying the nature of the noise (for instance, the noise can be Brownian or Poisson like). Moreover, unravelling the deterministic Lindblad equation in a stochastic differential equation (classical or quantum) offers some technical advantages (for instance in identifying the mean dynamics of the slow mode in the XXZ case).

4- Sorry. We corrected it.

5- We suppressed the Appendix G which contained extra information not explicitly used in the main text.

6- We corrected the misprints we found in the text.

Resubmission 1706.03984v4 (13 October 2017)
Resubmission 1706.03984v3 (7 September 2017)
Submission 1706.03984v2 (23 June 2017)

Invited Reports on this Submission

Toggle invited reports view

Anonymous Report 1 on 2017-9-25
Report

1)
The authors have improved the introduction by removing some of
the technical parts. In my opinion it has become more reader-friendly
after the revision.

3)
Despite the authors’ surprise, I believe that the question
has its right of existence. Indeed, the Langevin formalism is
by far much better founded and understood for classical than
for quantum systems. From Eq. (4) of the manuscript it does
not become clear to me, what is the freedom of choice (after
fixing the $e_j$ Lindblad operators) in the stochastic version
of the equation? What extra assumptions (not present in
the deterministic Lindblad formalism) are to be made on the
system-bath coupling and how do they yield different
Langevin-type equations? In their reply, the authors have
already given some keywords. However I would find it rather
important to include a short discussion about the construction
of quantum Langevin equations in the manuscript, since right
now there is very little information given, despite being
an object of central importance.

On the other hand, it is not true that the Lindblad formalism
only codes for mean quantities. In fact, there have been
successful attempts to calculate even large deviation functions
for spin chains (see e.g. PRL 112, 040602 (2014)) without
invoking the stochastic calculus method.

Requested changes

Include a better discussion of quantum Langevin equation

  • validity: high
  • significance: good
  • originality: high
  • clarity: good
  • formatting: good
  • grammar: excellent

---

## Round 4 · Author Response

1) Following the referee’s suggestion, we have rewritten the general discussion of quantum stochastic processes, using a more intuitive and physical approach.

2) Indeed, the author of ref. PRL 112, 040602 (2014) computed large deviation functions. These refer to quantum randomness encoded into a system quantum state (time evolving according to a Lindblad equation). As we said in our previous answer, there are two origins of randomness in stochastic quantum theory : that due to the noise and that due to the probabilistic nature of quantum mechanics. Ref. PRL 112, 040602 (2014) deals with those due to the probabilistic nature of quantum mechanics. In presence of external noise, as in the models we discussed in this paper, there are also randomness due to that noise. Those are not encoded into the mean system quantum state and hence not in the (mean) Lindblad equation, but in stochastic extensions of it which we describe in the paper.

---

## Editorial Decision

published